# PROBELLM: Automating Principled Diagnosis of LLM Failures

Yue Huang [* 1]   Zhengzhe Jiang [* 1]   Yuchen Ma [* 2]   Yu Jiang [1]   Xiangqi Wang [1]   Yujun Zhou [1]   Yuexing Hao [3]
Kehan Guo [1]   Pin-Yu Chen [4]   Marzyeh Ghassemi [3]   Stefan Feuerriegel [2]   Xiangliang Zhang [1]
 Toolkit    Docs

## Abstract

Understanding how and why large language models (LLMs) fail is becoming a central challenge as models rapidly evolve and static evaluations fall behind. While automated probing has been enabled by dynamic test generation, existing approaches often discover isolated failure cases, lack principled control over exploration, and provide limited insight into the underlying structure of model weaknesses. We propose PROBELLM, a benchmark-agnostic automated probing framework that elevates weakness discovery from individual failures to structured *failure modes*. PROBELLM formulates probing as a hierarchical Monte Carlo Tree Search, explicitly allocating limited probing budgets between global exploration of new failure regions and local refinement of recurring error patterns. By restricting probing to verifiable test cases and leveraging tool-augmented generation and verification, PROBELLM grounds failure discovery in reliable evidence. Discovered failures are further consolidated into interpretable failure modes via failure-aware embeddings and boundary-aware induction. Across diverse benchmarks and LLMs, PROBELLM reveals substantially broader, cleaner, and more fine-grained failure landscapes than static benchmarks and prior automated methods, supporting a shift from case-centric evaluation toward principled weakness discovery.

## 1. Introduction

Large language models (LLMs) are advancing at a pace that increasingly outstrips the evaluation ecosystems designed to assess them (Chang et al., 2023). While new benchmarks continue to emerge, most evaluations remain static snapshots of model behavior: prompts, tasks, and failure criteria are fixed in advance (Ni et al., 2025; Huang et al., 2025b). As models evolve, their error distributions shift accordingly, which makes it difficult for static evaluations to surface emerging weaknesses.

A natural response to these limitations has been to pursue active automation in evaluation and weakness discovery (Hou et al., 2025; Wang et al., 2025b; Lin et al., 2025; Cheng et al., 2024). Recent work has explored automated methods across red-teaming or adaptive probing pipelines that generate test cases on the fly to elicit model failures (Liu et al.; Huang et al., 2025b; Deng et al., 2024; Song et al., 2025). In parallel, another line of work has proposed dynamic benchmarks (Zhu et al.; Li et al., 2025; White et al., 2024), which update or expand evaluation datasets over time in an effort to better track evolving model behavior.

However, existing automated evaluations face several unresolved challenges. First, many automated probing pipelines are *not benchmark-agnostic* and require substantial *manual effort* to design data-construction procedures tailored to specific tasks, domains, or model capabilities (Zhu et al., 2024; Huang et al., 2025e; 2024c; Lin et al., 2025). Moreover, test generation is often driven by heuristic or loosely guided strategies that lack a *principled mechanism for exploration control*. Without explicit budget-aware allocation, probing effort may be wasted on redundant or low-yield tests, slowing the discovery of emerging weaknesses and limiting scalability. Second, most approaches remain *case-centric*, i.e., accumulating individual failure cases (Cheng et al., 2024; Rao et al., 2025), rather than identifying *failure modes*: recurring, structured patterns that explain how and why models fail across related inputs (as exemplified in Figure 1). Third, automated pipelines frequently rely on *unvalidated model-generated test cases* (Cheng et al., 2024). Ambiguous prompts, incorrect ground truths, or generation artifacts can be misinterpreted as genuine model failures, making it difficult to distinguish true model limitations from evaluation noise (Luo et al., 2025; Truong et al., 2025).

These limitations suggest that automating test generation alone is insufficient. What is needed is an automated prob-

[1]University of Notre Dame [2]LMU Munich [3]MIT [4]IBM Research. Correspondence to: Yue Huang <yhuang37@nd.edu>, Xiangliang Zhang <xzhang33@nd.edu>.

ing approach that explicitly manages exploration under budget constraints, elevates discovery from individual failure cases to interpretable failure modes, and grounds failure identification in verifiable evidence.

To meet these needs, we propose PROBELLM, an automated, *benchmark-agnostic*[1] probing framework for discovering and characterizing *failure modes*. PROBELLM formulates failure discovery as a hierarchical Monte Carlo Tree Search (MCTS) process with two complementary regimes: MACRO exploration steers probing toward under-explored regions to surface new failure areas, while MICRO refinement performs local perturbations around promising seeds to consolidate recurring patterns, enabling principled and efficient budget allocation. To reduce spurious failures, PROBELLM restricts to test cases with *verifiable* ground-truth answers and employs tool-augmented generation for grounded construction and verification. Finally, it converts individual failures into diagnostically meaningful modes via *failure-aware embeddings* and *boundary-aware induction*, summarizing each cluster with representative central and contrastive boundary cases to avoid over-generalization. We evaluate PROBELLM on five benchmarks and a diverse set of LLMs, analyzing its effectiveness, efficiency, and the quality of discovered failure modes. The results show that PROBELLM consistently uncovers broader and more fine-grained failure landscapes than static benchmarks and prior automated methods, while producing more diagnostically meaningful failure modes.

Overall, this paper makes **three contributions**:

1. We propose PROBELLM, a benchmark-agnostic automated probing approach for discovering and characterizing failure modes of large language models;

2. PROBELLM introduces a principled failure-mode discovery methodology that formulates probing as a hierarchical MCTS, balances global exploration and local refinement, grounds failures in verifiable test cases, and induces interpretable failure modes via failure-aware embeddings and boundary-aware induction;

3. Through extensive experiments across multiple datasets and models, we demonstrate the effectiveness of PROBELLM and provide insights for future studies.

## 2. Preliminaries

**Probing target.** Let $f_\theta$ denote the LLM under evaluation (target model). Our goal is to discover its *systematic failure modes*: structured patterns of incorrect behavior that reflect persistent model weaknesses rather than isolated errors.

---

[1]"Benchmark-agnostic" here means PROBELLM is a no-benchmark-specific construction pipeline that benchmarks are used only as an optional source of seed queries for initialization, and can be replaced by arbitrary user-provided prompts.

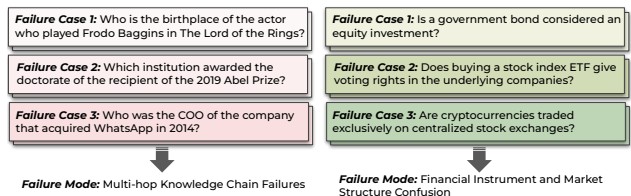

*Figure 1.* Examples of individual failure cases and the corresponding failure modes that capture recurring error patterns.

**Test cases and failures.** We consider a space of test cases $\mathcal{X}$, where each $x \in \mathcal{X}$ is an input prompt (with optional context). Querying the target model produces an output $y = f_\theta(x)$. Given a reference answer $y^\star(x)$ and a verifier $V : \mathcal{Y} \times \mathcal{Y} \to \{0, 1\}$ that returns 1 iff the output is verified as correct, we define the failure indicator:

$$\mathbb{I}_{\text{fail}}(x) \triangleq \mathbb{1}\big[\, V\big(f_\theta(x),\, y^\star(x)\big) = 0\,\big], \qquad (1)$$

where $\mathbb{I}_{\text{fail}}(x) = 1$ indicates that the verifier rejects the model output (a *failure case*).

**Adaptive probing.** An adaptive probing policy selects the next test case by conditioning on past outcomes. Given the history $\mathcal{H}_t = \{(x_i, \mathbb{I}_{\text{fail}}(x_i))\}_{i=1}^t$, it generates or selects

$$x_{t+1} \sim \pi(\cdot \mid \mathcal{H}_t), \qquad (2)$$

following a policy $\pi$. After $T$ probes, the set of failure cases is denoted as

$$\mathcal{F}_T = \{x_t \mid \mathbb{I}_{\text{fail}}(x_t) = 1,\ t = 1, \ldots, T\}. \qquad (3)$$

**From failure cases to failure modes.** Individual failure cases provide limited diagnostic insight. We therefore seek *failure modes*, which summarize recurring patterns of failures across related inputs, as illustrated in Figure 1.

> **Failure case.** A *failure case* is an individual test instance on which the target model produces an incorrect output.
>
> **Failure mode.** A *failure mode* is a recurring and structured pattern of failures, manifested as a coherent set of failure cases that share a common error mechanism or underlying cause, reflecting *how* the model fails rather than merely *what* the input is about.

Formally, a *failure mode* is a structured subset $\mathcal{M} \subseteq \mathcal{F}_T$ on which the model fails consistently. Failure mode synthesis can be viewed as a *structured set discovery* problem, searching over a hypothesis class $\mathcal{G} \subseteq 2^{\mathcal{F}_T}$ for subsets $\mathcal{M} \in \mathcal{G}$ that capture recurring, high-prevalence failures. In practice, this can be implemented by embedding failure cases into a latent space where similar weaknesses form coherent regions; obtaining a mode set $\mathcal{M}$ then amounts to identifying such a coherent region in the latent space.

**Budget and objective.** A principled probing policy $\pi$ should maximize the discovery of diverse, recurring failure modes (and their supporting failure cases) within a fixed

probing budget (e.g., the number of probing steps or times). Formally, given budget $T_{\max}$, our objective is to find an adaptive probing policy $\pi$ that selects test cases $\{x_t\}_{t=1}^T$ with $T \leq T_{\max}$, so as to maximize the expected utility of the discovered failure set $\mathcal{F}_T$ for failure-mode discovery:

$$\pi^\star \in \arg\max_\pi \; \mathbb{E}[U(\mathcal{F}_T)] \quad \text{s.t.} \quad T \leq T_{\max}. \quad (4)$$

Here $U(\mathcal{F}_T)$ rewards both (i) discovering many distinct failure modes and (ii) collecting sufficient supporting evidence for each mode. For example, letting $K_T$ be the number of discovered modes, $n_k$ the number of failures associated with mode $k$, and $m$ a per-mode cap, we can define $U(\mathcal{F}_T) \triangleq \sum_{k=1}^{K_T} \min\{n_k, m\}$, which encourages breadth (more modes) and depth (enough cases per mode). While the above objective is difficult to optimize directly, it informs the design of our search components and heuristics.

## 3. The Proposed PROBELLM

### 3.1. Overview

We propose PROBELLM to realize such an adaptive probing policy $\pi$. Specifically, we instantiate $\pi$ with a **hierarchical** Monte Carlo Tree Search (MCTS) (Coulom, 2006; Vien & Toussaint, 2015) that iteratively proposes and verifies test cases, and allocates search effort to regions that yield recurring failures. In this paper, we focus on test cases with well-defined ground-truth answers to improve the reliability of failure verification, and discuss the scope and potential extensions to open-ended settings in Appendix B.2.

### 3.2. Hierarchical MCTS Formulation

The hierarchical MCTS in PROBELLM is illustrated in Figure 2 and uses a two-level hierarchy to balance *global exploration* and *local refinement*. At the top level, it chooses between MACRO and MICRO, which determines the search strategy used to propose the next probe $x_{t+1}$ (summarized below). Conditioned on this top-level choice, the second level runs a standard MCTS to select, expand, and evaluate candidate test cases within the corresponding search tree.

> **MACRO (coverage).** Broad exploration to increase topical diversity and surface novel failure regions.
>
> **MICRO (refinement).** Local exploration around a seed/topic to densify evidence and reveal coherent failure patterns.

Next, we introduce the detailed MCTS procedure.

①  **MCTS structure and node representation.** PROBELLM maintains a global root node $r$ with two children corresponding to the top-level *search regimes*, $\rho_{\text{MACRO}}$ and $\rho_{\text{MICRO}}$. For each regime $\mathsf{s} \in \{\text{MACRO}, \text{MICRO}\}$, it maintains its own search tree rooted at $\rho_\mathsf{s}$. Within each search tree, a node $u$ corresponds to a *concrete evaluated test case*,

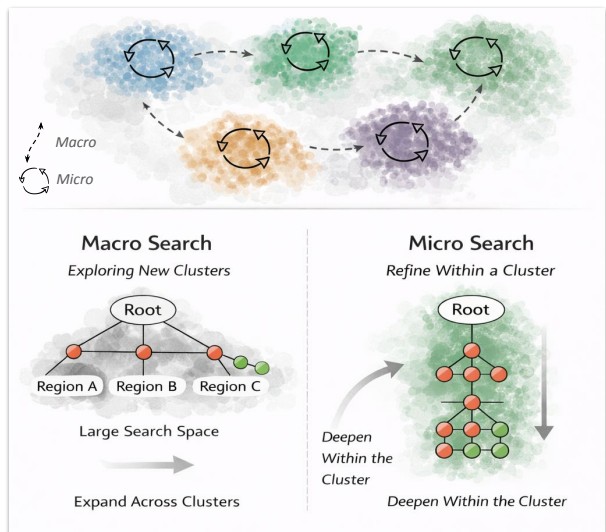

*Figure 2.* Differences of MACRO and MICRO search strategy. MACRO aims to diversify the search topics while MICRO aims to enhance local exploration.

represented as a tuple

$$u \equiv \big(x, y^\star(x), y(x), \mathbb{I}_{\text{fail}}(x)\big), \quad (5)$$

where $x \in \mathcal{X}$ is the question/prompt of test input, $y^\star(x) \in \mathcal{Y}$ is the verified reference (ground-truth) answer, $y(x) = f_\theta(x)$ is the target model output, and $\mathbb{I}_{\text{fail}}(x)$ indicates whether the model $f_\theta$ fails on $x$. Expanding a node $u$ proposes child test cases conditioned on the current one. Under MACRO, the proposal mechanism diversifies $x$ of $u$ to explore broadly; under MICRO, it generates targeted variants around $x$ to densify evidence within a suspected failure region. Each proposed child is then evaluated by querying $f_\theta$ and $V$ to populate its stored tuple.

②  **Node statistics.** Following standard MCTS bookkeeping, each node $u$ tracks a visit count $N(u)$; in our setting, a visit corresponds to expanding $u$ by proposing and evaluating a child test case. We additionally track $E(u)$, the number of these evaluated children that are failure cases. We use these statistics to guide tree search toward nodes that both (i) tend to yield failures when expanded and (ii) remain under-explored. In particular, we estimate the empirical failure rate of *expansions from* $u$ by

$$\widehat{p}(u) \triangleq \frac{E(u)}{\max\{1, N(u)\}}. \quad (6)$$

This approximates the probability that a newly proposed child conditioned on $u$ becomes a failure case.

③  **Node selection via UCB.** During tree traversal, we select which node to expand using a UCB-style rule that balances exploiting nodes with high empirical failure rates and exploring less-visited nodes. Concretely, for a parent node $u$

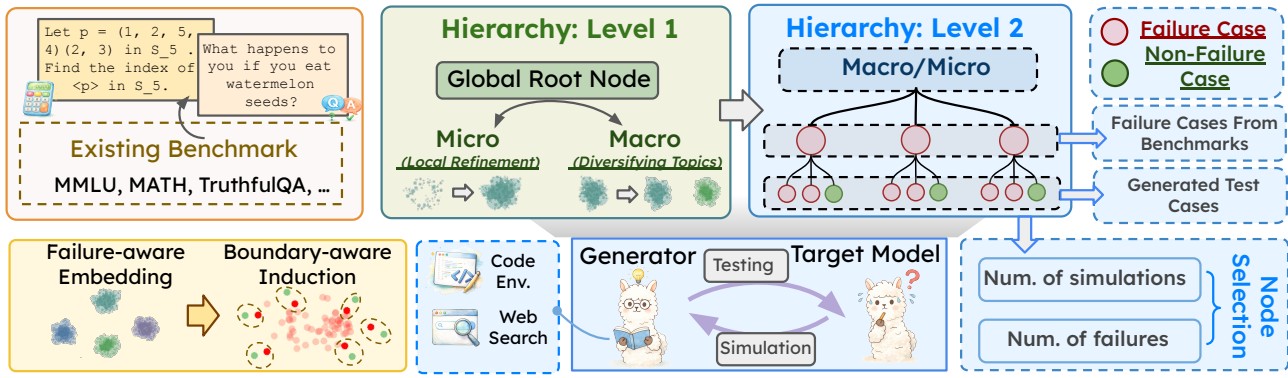

*Figure 3.* **Overview of PROBELLM.** (I) PROBELLM probes a target model using an LLM-based generator and initializes the search with seed test cases from an existing benchmark. (II) At Level 1, the hierarchical search selects between MACRO and MICRO regimes. Conditioned on the selected regime, PROBELLM performs tool-augmented generation to propose new test cases and verifies the target model responses. (III) From the collected failure cases, PROBELLM computes failure-aware embeddings, clusters failures, and applies boundary-aware induction to produce interpretable failure modes.

and a candidate child $v \in \mathrm{Ch}(u)$, we define

$$\mathrm{UCB}(v \mid u) \triangleq \widehat{p}(v) + \beta \sqrt{\frac{\log(\max\{1, N(u)\})}{\max\{1, N(v)\}}}, \quad (7)$$

where $\beta > 0$ controls the exploration–exploitation trade-off. We then choose

$$v^\star \in \arg \max_{v \in \mathrm{Ch}(u)} \mathrm{UCB}(v \mid u), \quad (8)$$

and set $u \leftarrow v^\star$ to continue traversal. We continue this selection step until reaching a node $\tilde{u}$ that is *expandable*, i.e., it has not reached the maximum width: $|\mathrm{Ch}(\tilde{u})| < W_{\max}$.

④ **Expansion via strategy-conditioned generation.** When expanding a node $u$ under search strategy $\mathsf{s} \in \{\mathrm{MACRO}, \mathrm{MICRO}\}$, PROBELLM generates a new candidate test case

$$x_{\mathrm{new}} \sim G_{\mathsf{s}}(\cdot \mid u), \quad (9)$$

where $G_{\mathrm{MACRO}}$ favors topical diversity and coverage, and $G_{\mathrm{MICRO}}$ favors local perturbations and neighborhood exploration around the selected context. More details are introduced in section 3.3.

⑤ **Simulation and backup.** Given $x_{\mathrm{new}}$, we query the target model and verify correctness: $\mathbb{I}_{\mathrm{fail}}(x_{\mathrm{new}}) = \mathbb{1}[V(f_\theta(x_{\mathrm{new}}), y^\star(x_{\mathrm{new}})) = 0]$. If $\mathbb{I}_{\mathrm{fail}}(x_{\mathrm{new}}) = 1$, we add $x_{\mathrm{new}}$ to $\mathcal{F}$. Let $\mathcal{P}(x_{\mathrm{new}})$ denote the set of nodes on the selected path, including the chosen search strategy root. We then update every node on the path

$$N(u) \leftarrow N(u) + 1, \ E(u) \leftarrow E(u) + \mathbb{I}_{\mathrm{fail}}(x_{\mathrm{new}}). \quad (10)$$

### 3.3. Tool-Augmented Generation $G_{\mathsf{s}}$

In PROBELLM, $G_{\mathsf{s}}$ is implemented as a tool-augmented LLM $g_\psi$, which is equipped with a set of tools $\mathcal{T}$, such as web retrieval and a Python execution environment. These tools support question construction and also help obtain or verify the ground-truth answer $y^\star(x_{\mathrm{new}})$.

**MACRO generator $G_{\mathrm{MACRO}}$.** To expand coverage, $G_{\mathrm{MACRO}}$ steers generation toward topics that are under-represented among previously explored nodes. To operationalize "under-represented," we summarize the current search frontier in an embedding space. Let $\mathrm{emb} : \mathcal{X} \to \mathbb{R}^d$ be an embedding function, and $z(u)$ be the embedding of $x$ in node $u$. We periodically cluster the set of embeddings $\{z(u)\}$ into $K$ clusters, where $K$ is a user-specified number (denoted $n$ in our implementation). For each cluster $k \in \{1, \ldots, K\}$ with centroid $\mu_k$, we select a representative node (medoid)

$$u_k \in \arg \min_{u \in \mathcal{C}_k} \|z(u) - \mu_k\|_2, \quad (11)$$

and form a representative set $\mathcal{R} = \{u_1, \ldots, u_K\}$. We then provide $\mathcal{R}$ as context to the generator $g_\psi$ and instruct it to propose $x_{\mathrm{new}}$ that is *topically distinct* from these representatives, thereby encouraging exploration beyond the currently covered regions. During generation, tool calls (e.g., retrieval) are allowed to support factual grounding and verification.

**MICRO generator $G_{\mathrm{MICRO}}$** intensifies search within the local neighborhood of a selected node $u$. Specifically, given $x$ on node $u$, the generator produces $x_{\mathrm{new}}$ via controlled perturbations of $x$ (e.g., entity or attribute substitutions) to stay within the same underlying topic while varying surface realizations. This local refinement densifies failure evidence around the seed and helps delineate coherent error patterns

that correspond to failure modes. The generator may invoke tools in $\mathcal{T}$ to obtain or verify $y^\star(x_{\text{new}})$.

### 3.4. Synthesizing Failure Cases into Failure Modes

After collecting a set of failure cases $\mathcal{F}_T$, PROBELLM consolidates them into a smaller set of interpretable *failure modes*. Recall that a failure mode is a structured subset $\mathcal{M} \subseteq \mathcal{F}_T$ on which the model fails consistently. We synthesize modes by embedding failure cases into a latent space where similar weaknesses form coherent regions; a mode corresponds to such a region mapped back to a subset of $\mathcal{F}_T$. Concretely, we compute failure-aware embeddings, identify coherent regions by clustering, and induce concise descriptions of $\mathcal{M}$ with LLM-assisted inductions.

**Step I: Failure-aware embeddings.** A key requirement for failure-mode synthesis is that grouping should reflect *how* the model fails, not just *what* the input is about. Embeddings based only on the prompt semantics tend to cluster by topic/domain, which conflates distinct error mechanisms within the same task. We therefore use *failure-aware* embeddings that incorporate failure signals (e.g., the model output and verifier outcome), so that proximity in embedding space better aligns with shared failure patterns.

Concretely, for each failure case, we concatenate the problem description with an error description derived from the mismatch between the model output $y(x)$ and the ground-truth answer $y^\star(x)$ using an LLM (i.e., GPT-5.2). An embedding model maps this combined signal to $z(x) = \text{emb}(x \,\|\, \text{error}(x)) \in \mathbb{R}^d$, so that similarity in the embedding space reflects both semantic context and failure characteristics[2].

**Step II: Clustering and boundary-aware induction.** We group failure embeddings $\{z(x) \mid x \in \mathcal{F}\}$ into $\mathcal{C}_1, \ldots, \mathcal{C}_K$ using HDBSCAN (McInnes et al., 2017), which identifies dense regions and marks outliers as noise (clustering sensitivity is analyzed in Section 5). Each dense region forms a cluster, which may represent a candidate failure mode, but still include mixed multiple error mechanisms or reflects spurious correlations. To identify a common failure mode, we must extract a shared weakness pattern and delimit *where* it applies, using nearby verified *non-failures* as contrast.

This motivates a *boundary-aware* induction strategy that explicitly surfaces *where the failures stop*[3]. For each cluster $\mathcal{C}_k$, we construct an *information-maximizing* evidence set consisting of (i) *central* failures that capture the dominant error pattern, and (ii) *boundary* failures paired with their nearest verified *non-failures*, which contrast failure and success

---

[2]We use text-embedding-3-small as the embedding model.

[3]We use "boundary" in an empirical, diagnostic sense, not as a formal classifier decision boundary.

under minimally different conditions and expose the failure decision boundary. Concretely, for a cluster $\mathcal{C}_k$ we select a small set of $n_c$ (e.g., 3) central members $\mathcal{S}_k^{\text{core}} \subset \mathcal{C}_k$ (i.e., closest to the centroid $\mu_k$) to elicit the common error mechanism. We also identify $b$ boundary members $\mathcal{S}_k^{\text{bd}} \subset \mathcal{C}_k$ (i.e., peripheral points in embedding space). For each boundary failure $x \in \mathcal{S}_k^{\text{bd}}$, we retrieve a nearest verified non-failure $x^+$ from the search tree. We then prompt an LLM with $\mathcal{S}_k^{\text{core}}$ together with the contrastive pairs $\{(x, x^+) | x \in \mathcal{S}_k^{\text{bd}}\}$ to induce a concise natural-language description of the underlying failure mode for $\mathcal{C}_k$.

We show the full algorithm of PROBELLM in Appendix B, and theoretical analysis in Appendix E.

## 4. Implementation Details

To improve clarity and reproducibility without overloading the draft, we defer detailed implementation choices to the codebase and documents. This includes the prompt templates used for probing, as well as concrete hyperparameter settings and other engineering-level configurations.

## 5. Experiments

### 5.1. Experimental Setup

**Model & Benchmark Selection.** We evaluate 12 target models covering open-weight and proprietary ones (more detailed in Appendix C). We use three generators to answer research questions (GPT-5.2 for RQ1 and RQ2, Llama-3.1-70B-Ins., claude-4.5-sonnet for RQ2). For the initial benchmarks, we select five benchmarks with different evaluation aspects: MMLU (Hendrycks et al., 2020), SuperGLUE (Wang et al., 2019), MBPP (Austin et al., 2021), HellaSwag (Zellers et al., 2019), and TruthfulQA. More details are presented in Appendix C.

**Evaluation Protocols & Metrics.** We employ complementary evaluation protocols to assess both *failure exposure* and *failure-mode discovery*. **(a)** For the direct evaluation of PROBELLM, we report the **error rate**, defined as the percentage of generated questions that are answered incorrectly, to measure efficiency under a fixed budget. Beyond aggregate failure frequency, we further assess the *structure and novelty* of the discovered failure modes. Specifically, we report the **noise rate** and the **cluster-size standard deviation** to evaluate the stability and compactness of failure-mode clusters. Moreover, to directly measure *novel failure-mode discovery*, we adopt a **cluster-overlap analysis** (RQ4) that compares clusters induced from PROBELLM with those derived from existing static benchmarks, where lower overlap indicates higher mode-level novelty. Formal definitions of all metrics are provided in Appendix C. **(b)** For human evaluation, we go beyond aggregate error rates and explicitly

*Table 1.* Main results across 12 target models, covering both proprietary models (light blue) and open-weight models (light purple). Proportions of MACRO (MA.) and MICRO (MI.) simulations, and the error rate (%) are reported.

| Target Model | MMLU | | | SuperBLUE | | | MBPP | | | HellaSwag | | | TruthfulQA | | |
|---|---|---|---|---|---|---|---|---|---|---|---|---|---|---|---|
| | Sim. Type | | Error | Sim. Type | | Error | Sim. Type | | Error | Sim. Type | | Error | Sim. Type | | Error |
| | MA. | MI. | Rate$_{(\%)}$ | MA. | MI. | Rate$_{(\%)}$ | MA. | MI. | Rate$_{(\%)}$ | MA. | MI. | Rate$_{(\%)}$ | MA. | MI. | Rate$_{(\%)}$ |
| Grok-4.1-fast | 34% | 66% | 47% | 50% | 50% | 39% | 35% | 65% | 39% | 69% | 31% | 35% | 45% | 55% | 39% |
| Gemini-2.5-flash | 63% | 37% | 38% | 45% | 55% | 46% | 53% | 47% | 54% | 76% | 24% | 43% | 56% | 44% | 38% |
| Claude-3.5-sonnet | 36% | 64% | 68% | 53% | 47% | 50% | 42% | 58% | 62% | 76% | 24% | 59% | 76% | 24% | 54% |
| GPT-4o-mini | 54% | 46% | 65% | 53% | 47% | 65% | 31% | 69% | 76% | 62% | 38% | 71% | 47% | 53% | 56% |
| devstral | 40% | 60% | 66% | 59% | 41% | 44% | 27% | 73% | 50% | 61% | 39% | 45% | 66% | 34% | 47% |
| Llama-3.1-8b-ins. | 59% | 41% | 86% | 46% | 54% | 71% | 40% | 60% | 90% | 62% | 38% | 75% | 62% | 38% | 78% |
| phi-4 | 64% | 36% | 70% | 58% | 42% | 59% | 39% | 61% | 78% | 75% | 25% | 52% | 53% | 47% | 69% |
| Deepseek-v3.2 | 51% | 49% | 36% | 34% | 66% | 43% | 65% | 35% | 46% | 71% | 29% | 41% | 57% | 43% | 43% |
| ministral-14b | 37% | 63% | 70% | 48% | 52% | 65% | 42% | 58% | 64% | 82% | 18% | 62% | 47% | 53% | 64% |
| olmo-3-7b-ins. | 62% | 38% | 73% | 58% | 42% | 69% | 59% | 41% | 76% | 73% | 27% | 64% | 81% | 19% | 64% |
| granite-4.0 | 51% | 49% | 72% | 61% | 39% | 75% | 50% | 50% | 83% | 75% | 25% | 68% | 58% | 42% | 70% |
| GPT-oss-20b | 73% | 27% | 36% | 48% | 52% | 31% | 68% | 32% | 32% | 56% | 44% | 46% | 45% | 55% | 40% |

*Table 2.* Cluster statistics between existing benchmarks and PRO-BELLM. Avg. size means the average cluster size.

| Target Model | Existing Bench | | PROBELLM | |
|---|---|---|---|---|
| | #Cluster | Avg. Size | #Cluster | Avg. Size |
| Deepseek-v3.2 | 4 | 30.25 | 10 | 10.80 |
| Llama-3.1-8b-ins. | 4 | 60.25 | 19 | 12.58 |
| Claude-3.5-sonnet | 2 | 114.00 | 11 | 13.55 |
| Ministral-14b | 6 | 25.00 | 18 | 10.61 |

*Table 3.* Noise rate (**NR**) and cluster-size standard deviation (**Std**$_{CS}$) across methods and models.

| Method | Baseline | NR ↓ | Std$_{CS}$ ↓ |
|---|---|---|---|
| **AutoDetect** | llama-3.1-8b-ins. | 60.9 | 13.2 |
| **PAIR** | llama-3.1-8b-ins. | 77.2 | 22.9 |
| | Claude-3.5-sonnet | 50.5 | 9.2 |
| | GPT-4o-mini | 45.6 | 8.8 |
| **PROBELLM** | llama-3.1-8b-ins. | 37.0 | 4.2 |
| | ministral-14b-2512 | 37.5 | 3.2 |
| | Grok-4.1-fast | 31.7 | 4.3 |

audit the quality of synthesized test cases and ground-truth answers, the reliability of the LLM-based verifier, and the coherence and diagnostic usefulness of the induced failure modes. Detailed protocols and metrics are provided in Appendix F. **(c)** For automatic verification, we use an LLM-as-a-Judge (GPT-5.2) (Zheng et al., 2023) to compare model responses against verified ground-truth answers.

### 5.2. Main Results

**RQ1** **How effective is PROBELLM at exposing failures across different seed benchmarks and target LLMs?**

From Table 1, we observe that PROBELLM consistently exposes a high fraction of model errors across all five datasets, demonstrating stable failure-discovery capability across tasks and domains. Notably, high error rates are observed for both proprietary and open-weight models, suggesting that the discovered failures reflect intrinsic model weaknesses rather than model accessibility. In addition, the proportions of MACRO and MICRO simulations are generally balanced across datasets and models, indicating that the search process is not biased toward a specific simulation type. As shown in Figure 6, the number of discovered failure clusters is strongly correlated with the average error rate (Pearson $r = 0.864$), implying that models exhibiting higher error rates tend to have more diverse failure modes. This suggests that cluster count serves as a meaningful signal of failure diversity rather than an artifact of the discovery process.

**RQ2** **How do search depth and tool usage in MCTS affect PROBELLM's effectiveness?**

To assess how MCTS design affects PROBELLM, we first study whether increasing search depth improves failure discovery. As shown in Figure 4, across 12 target models, increasing the search depth improves the discovered failure rate on four of the five datasets; on SuperGLUE, the slight drop at depth 4 likely reflects saturation, where additional search yields redundant or less informative cases. When tools are used in the expansion of MCTS, Figure 5 shows that testcase generation is dominated by Python execution, while answer generation uses a more balanced mix of Python execution and web search. Despite differing usage frequencies, tool-augmented interactions expose failures more effectively than perturbation, suggesting that deeper reasoning and external tool grounding better surface model weaknesses. We also analysis the impact of different generators in Appendix C.4. We further analyze the relationship between the MACRO/MICRO simulation ratio (i.e., $\beta$ in Eq 7) and the error rate, and observe no monotonic or fixed correlation, indicating that PROBELLM's effectiveness does not rely on a specific allocation but on adaptive hierarchical search (see Appendix C.5 for details).

**RQ3** **How valid are PROBELLM's synthesized test**

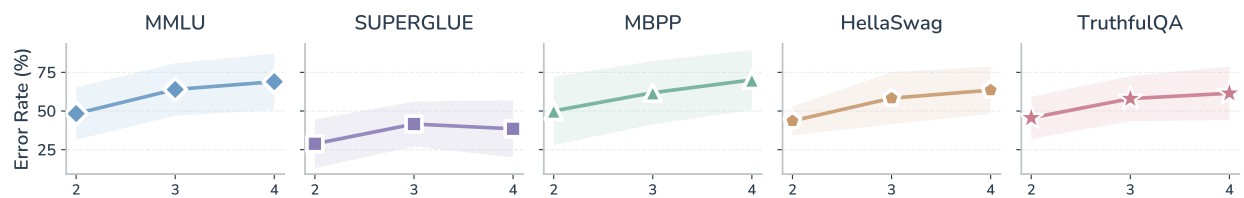

*Figure 4.* Error rates at different search depths across five datasets, with standard deviations computed over results from 12 target models.

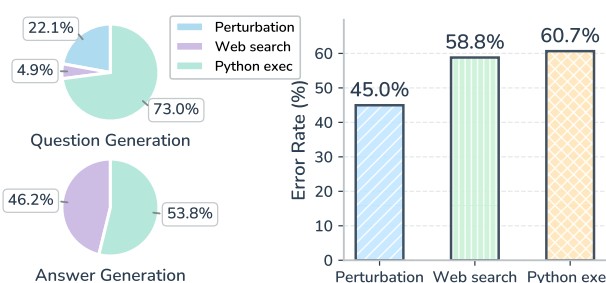

*Figure 5.* Tool usage distribution for question generation (top pie) and answer generation (bottom pie), along with error rates across tools (right bar chart).

*Table 4.* Baseline comparison (error rate (%)) across different datasets (with `GPT-5.2` as the generator and `Llama-3.1-8b-ins.` as the target model). We also report the final attack success rate of PAIR in the subscripts.

| Dataset | AutoDetect | PAIR | PROBELLM |
|---------|-----------|------|----------|
| Hella.  | 45%       | 35% $_{99.06\%}$ | **75%** |
| MBPP    | 46%       | 38% $_{94.85\%}$ | **90%** |
| MMLU    | 67%       | 46% $_{94.87\%}$ | **86%** |
| Super.  | 38%       | 44% $_{89.50\%}$ | **71%** |
| Truth.  | 56%       | 40% $_{93.86\%}$ | **78%** |

cases, automatic verification, and induced failure modes according to human judgments?

We report the human evaluation results in Appendix C.3, which confirm the quality of synthesized test cases, the reliability of automatic verification, and the coherence and diagnostic usefulness of the induced failure modes.

**RQ4** **How is PROBELLM better than static benchmarks for coverage and novel failure-mode discovery?**

Table 2 and Figure 8 compare PROBELLM with existing static benchmarks in terms of coverage and the discovery of failure modes. Across all target models, PROBELLM consistently identifies substantially more failure clusters while yielding much smaller average cluster sizes, indicating finer-grained and more diverse failure modes rather than coarse aggregation. To fairly assess overlap, we construct a balanced setting by mixing an equal number of failure cases discovered by PROBELLM and those from existing benchmarks, and then perform clustering on the combined set. The resulting Venn diagrams show that only a small fraction of clusters overlap, while a large number of clusters

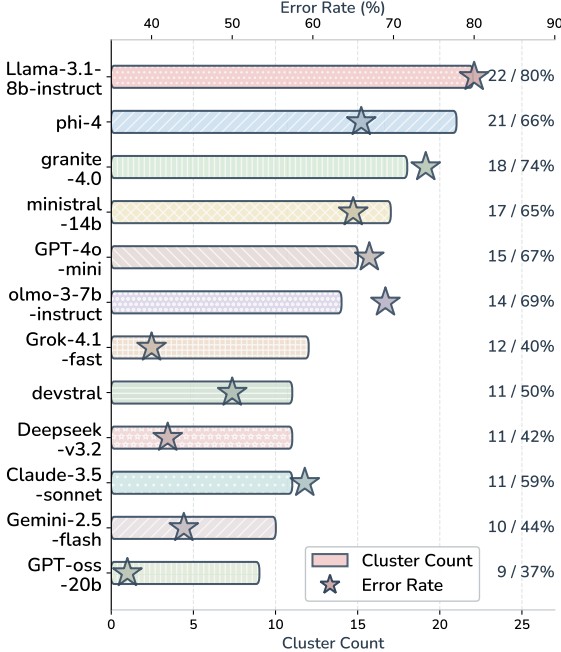

*Figure 6.* Cluster count and average error rate of different target models (with Pearson $r = 0.864$).

are discovered exclusively by PROBELLM. These results demonstrate that PROBELLM not only covers most failure modes identified by static benchmarks, but also uncovers a substantial set of previously unseen failure modes, significantly expanding evaluation coverage.

**RQ5** **How does PROBELLM perform relative to prior automated probing and evaluation methods?**

We also compare PROBELLM with two representative prior baselines: AutoDetect (Cheng et al., 2024) and PAIR (Chao et al., 2025) (the implementation details are shown in Appendix D). Table 6 shows that PROBELLM consistently outperforms AutoDetect and PAIR across all five datasets, achieving substantially higher error rates under the same generator–target setting. Notably, although PAIR attains high final attack success rates, its induced error rates remain much lower, indicating that high attack success does not necessarily imply stronger failure discovery.

Moreover, as shown in Table 3, both baselines exhibit high noise rates and large cluster-size variability, indicating spurious discoveries (AutoDetect) or poorly structured failure modes (PAIR). In contrast, PROBELLM consistently

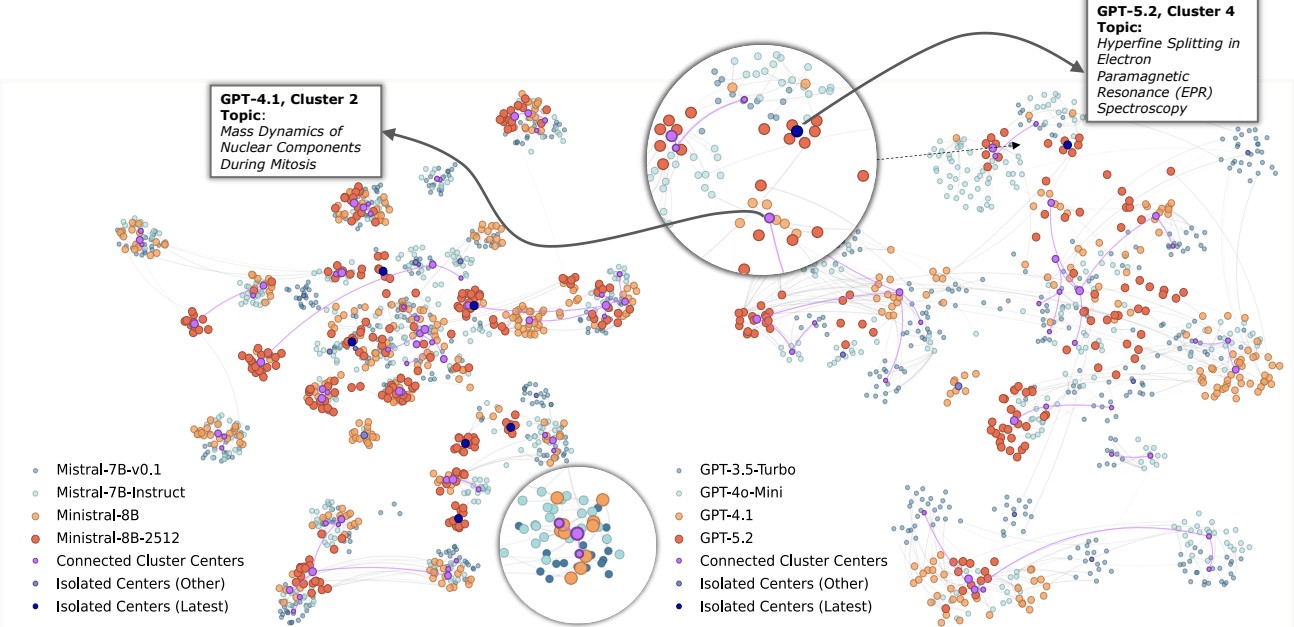

*Figure 7.* UMAP visualization of failure samples identified by PROBELLM for Mistral (left) and GPT (right) families. Small nodes represent individual failed queries, colored by version (Cool: Older → Warm: Newer). Directed edges connect semantically similar failures across generations, tracing the evolution of model deficits. Large purple centroids indicate weakness clusters persisting across versions, while dark blue centroids are isolated. Insets highlight dense failure concentrations in specific scientific domains (e.g., Nuclear Dynamics), illustrating how error patterns morph structurally even in advanced models.

achieves lower noise rates and reduced cluster-size standard deviation across target models, suggesting cleaner discoveries and more balanced failure-mode structures, even on stronger models. We also analysis the quality of generated cases (questions) by human evaluation in Appendix C.2.

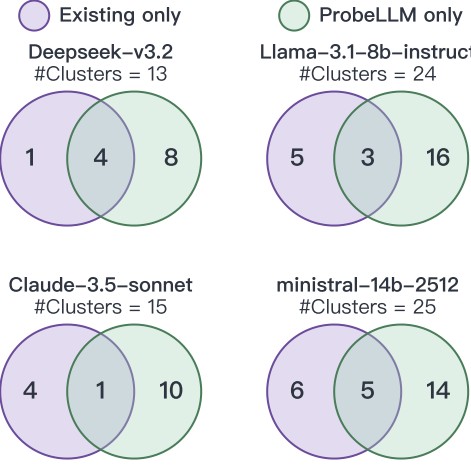

*Figure 8.* Failure modes discovered by existing benchmarks and PROBELLM. Overlapping regions indicate failure modes that contain failure cases identified by both existing benchmarks and PROBELLM.

**RQ6** **Can the failures discovered by PROBELLM be leveraged to improve model performance through mitigation or fine-tuning?**

As shown in Figure 9, across both models, error-focused training consistently improves performance on the error-prone evaluation set, with gains increasing up to moderate training sizes (150–200 samples) and diminishing thereafter. The slight performance drop observed at larger scales is potentially attributed to over-concentration on narrow failure distributions, where redundant failure cases bias learning toward local patterns. Importantly, performance on the general evaluation set remains stable across all settings, indicating that the proposed training strategy enables targeted failure repair without degrading general benchmark performance (detailed experiment setting is shown in Appendix C.6).

**RQ7** **What is the computational and token cost of PROBELLM, and how does its cost-effectiveness compare across settings?**

In terms of cost efficiency, Table 5 indicates that employing GPT-5.2 as the generator incurs consistently low computational costs across all five benchmarks. For every dataset, the total token usage stays below 6K, and the highest cost required to produce a high–error-rate test case is less than $0.10. These results suggest that PROBELLM enables effective failure discovery while maintaining minimal monetary overhead, highlighting its practicality and cost-effectiveness.

**RQ8** **How do failure modes evolve across Mistral and GPT model generations?**

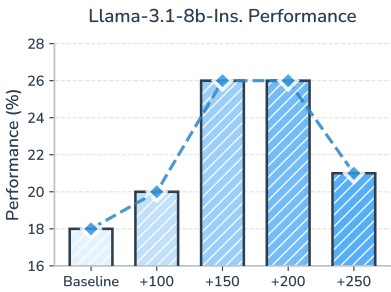 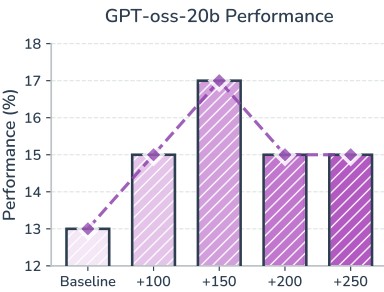 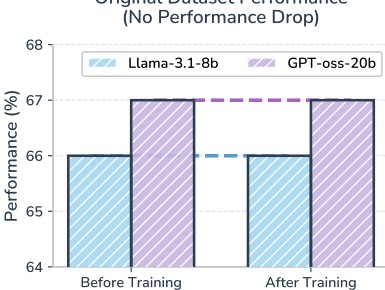

*Figure 9.* Performance of error-focused training with PROBELLM-discovered data under different training sizes. Left and middle panels report results on error-prone evaluation sets across five datasets, and the right panel reports performance on held-out subsets of existing benchmarks.

*Table 5.* Average cost breakdown by datasets (**Bench.**) and search type (**Type**: MA./MI.) across pipeline components, with GPT-5.2 as the generator. **Model$_{In/Out}$** is the model input/output cost; **Judge.** denotes the cost of judging/verification; **Tool$_Q$** and **Tool$_A$** are tool costs incurred during question generation and answer-stage processing. Superscripts in **Total** report the max corresponding average dollar cost (USD).

| Benchmark | Type | Model$_{In/Out}$ | Judge. | Tool$_Q$ | Tool$_A$ | Total |
|---|---|---|---|---|---|---|
| HellaSwag | MA. | 1210.04 | 1430.55 | 1666.34 | 553.18 | 4860.11$^{\$0.083}$ |
| | MI. | 985.64 | 850.46 | 600.04 | 415.63 | 2851.77$^{\$0.048}$ |
| MBPP | MA. | 1334.09 | 1480.21 | 1726.27 | 518.32 | 5058.89$^{\$0.086}$ |
| | MI. | 2190.26 | 1642.30 | 525.60 | 470.23 | 4828.39$^{\$0.082}$ |
| MMLU | MA. | 1276.75 | 1415.03 | 1572.67 | 546.84 | 4811.29$^{\$0.082}$ |
| | MI. | 2963.53 | 1608.42 | 538.37 | 582.68 | 5693.00$^{\$0.097}$ |
| SuperGLUE | MA. | 1184.78 | 1359.38 | 1510.34 | 556.26 | 4610.76$^{\$0.078}$ |
| | MI. | 1261.92 | 805.27 | 554.88 | 365.59 | 2987.66$^{\$0.051}$ |
| TruthfulQA | MA. | 1156.43 | 1442.13 | 1667.47 | 580.47 | 4846.50$^{\$0.082}$ |
| | MI. | 1462.49 | 946.24 | 435.76 | 380.31 | 3224.80$^{\$0.055}$ |

Figure 7 visualizes the evolution of failure modes for `Mistral` and `GPT` families, revealing trends of *convergence* and *specialization*. As models evolve, the failure landscape shifts from diffuse distributions to fewer, tighter clusters. Connected centroids trace how persistent semantic weaknesses are inherited and morphed across versions. In contrast, isolated centroids in advanced models highlight the retreat of deficits into niche domains. For example, the isolated cluster for "Hyperfine Splitting" illustrates this shift toward extreme specialization. Detailed analysis is provided in Appendix C.7.

**RQ9** **How sensitive is the clustering process in PROBELLM to hyperparameter choices?**

We observe consistently high cluster stability across a wide range of clustering settings, indicating that the discovered failure modes are robust (see Appendix C.8 for details).

## 6. Conclusion

We presented PROBELLM, which advances weakness discovery from isolated cases to structured failure modes via hierarchical Monte Carlo Tree Search. Grounded in verifiable evidence, PROBELLM reveals substantially broader failure landscapes than static benchmarks, underscoring the necessity of dynamic, mode-centric evaluation for the reliable characterization of evolving models.

## Impact Statement

This work develops an automated framework for adaptively probing large language models to discover recurring, systematic failure modes and induce them for debugging and monitoring. The primary positive impact is improving reliability, transparency, and accountability of LLM-based systems by moving beyond static benchmarks toward systematic, interpretable characterization of weaknesses that can guide mitigation and safer deployment. However, the same capability is dual-use: automated probing and tool-assisted test generation can be misused to more efficiently identify exploitable behaviors or craft adversarial prompts, and evidence gathered from external sources may introduce privacy, provenance, or bias concerns if raw traces are shared. To reduce these risks, we recommend restricting use to authorized evaluation settings, avoiding release of high-impact exploit prompts and sensitive retrieval traces (prefer aggregated failure-mode summaries), applying redaction and source controls, and using targeted human review to validate correctness and limit harmful content exposure. Finally, adaptive probing can increase compute cost; responsible deployment should emphasize efficiency, bounded budgets, and reuse of discovered failure modes to limit environmental footprint.

## Disclosure of Generative AI Usage

We used large language models to assist with writing clarity, grammar, and experimental code development, and used a text-to-image model for non-substantive visual elements. All final wording, technical claims, code, results, figures' scientific content, and conclusions were reviewed, validated, and produced by the authors.

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

# A. Related Work

**Trustworthy LLMs.** While large language models (LLMs) demonstrate remarkable capabilities, they also introduce a broad range of potential risks and threat vectors (Huang et al., 2025b; 2024b), including jailbreak attacks (Yi et al., 2024), privacy leakage (Kim et al., 2023), ethical violations (Jiao et al., 2025), and hallucinated responses (Huang et al., 2025a). Zou et al. (2023) propose the GCG method, which achieves universal and transferable jailbreak attacks through prompt prefix optimization. AutoDAN (Liu et al.) further introduces an automated framework for generating stealthy and semantically meaningful jailbreak prompts that can bypass alignment safeguards in LLMs. Beyond security vulnerabilities, several studies reveal systemic biases in LLMs across diverse downstream applications, as demonstrated by Dai et al. (2024), Yeh et al. (2023), and Ye et al. (2024). In multilingual settings, Huang et al. (2024a) identify inconsistencies and hallucinations in LLM outputs and propose a language aggregation strategy to alleviate these issues. Additionally, Berglund et al. uncover the "reversal curse," highlighting a fundamental limitation in LLM generalization. Prior work also shows that language models continue to struggle with moral reasoning and ethical dilemmas (Scherrer et al., 2023; Forbes et al., 2020). Moreover, recent studies have begun to expose emerging risks associated with LLM-based agents, including safety, controllability, and alignment concerns (Zhang et al., 2024c; Wang et al., 2025a; Huang et al., 2025d).

*Table 6.* Comparison with previous works across different dimensions. **Failure Mode Discovery** indicates whether a framework is capable of identifying structured failure modes (i.e., systematic patterns of model weaknesses), rather than merely collecting individual error or failure cases. **Data Quality Control** refers to the extent to which the generated test cases are ensured to be truthful, valid, and semantically consistent with the intended evaluation objectives. **Human Intervention** measures whether the evaluation process requires explicit manual intervention or curation, with methods requiring no human intervention marked higher. **Benchmark-Agnostic** denotes whether the evaluation is independent of fixed, publicly available benchmarks. **Failure Diversity** captures whether the framework explicitly enables coverage or analysis over diverse failure modes or regions of the input space.

| Framework | Failure Mode Discovery | Data Quality Control | *w/o* Human Intervention | Benchmark -Agnostic | Failure Diversity |
|---|---|---|---|---|---|
| ACE (Afkanpour et al., 2025) | ◑ | ✔ | ✘ | ✔ | ◑ |
| AutoDetect (Cheng et al., 2024) | ✔ | ◑ | ◑ | ✔ | ◑ |
| AutoBencher (Li et al., 2025) | ◑ | ✔ | ✔ | ✘ | ◑ |
| Adaptive Dis. (Wang et al., 2025b) | ✘ | ◑ | ✔ | ✔ | ✘ |
| FACT-AUDIT (Lin et al., 2025) | ◑ | ✔ | ◑ | ✔ | ✔ |
| APT (Rao et al., 2025) | ◑ | ◑ | ✔ | ✔ | ◑ |
| PROBELLM (Ours) | ✔ | ✔ | ✔ | ✔ | ✔ |

Legend: ✔ high/yes    ◑ partial/medium    ✘ low/no

**Probing for Weaknesses of LLMs.** A growing body of work focuses on systematically probing and characterizing the vulnerabilities of large language models. Cheng et al. (2024) propose AutoDetect, an automated framework that discovers hidden weaknesses in LLMs and leverages them to improve model robustness and overall performance. Followed by that, Li et al. (2025) and Bao et al. (2024) propose the frameworks for automatic benchmarking of language-related capabilities for generative models. ACE is an automated, active-learning framework for fine-grained and efficient evaluation of foundation model capabilities beyond static benchmarks (Afkanpour et al., 2025). SEAL (Zweiger et al., 2025) enables large language models to self-adapt their parameters to new tasks and knowledge. Qiu et al. (2023) introduce a benchmark for evaluating text safety and output robustness by assessing models' susceptibility to latent jailbreaks induced by subtle prompt perturbations. In the domain of mathematical reasoning, Hou et al. (2025) develop an automatic stress-testing framework that generates semantically equivalent yet more challenging problem variants to expose robustness failures. Wang et al. investigate contextual robustness by using automated tree search to generate adaptive distracting contexts, demonstrating that LLM reasoning can degrade significantly under irrelevant but semantically related information. Extending robustness analysis to multilingual settings, Xu et al. (2025) present an automatic probing approach that uncovers cross-lingual weaknesses in multilingual LLMs, revealing notable performance drops across languages. Beyond static evaluations, Liu et al. (2025) propose an online self-play reinforcement learning framework in which attacker and defender policies co-evolve, enabling models to adapt to dynamically evolving safety threats. Similarly, Rahman et al. (2025) introduce X-teaming, a multi-agent framework that models multi-turn jailbreak attacks and defenses as adaptive interactions, uncovering complex failure modes that are not captured by single-turn evaluations. Moreover, some recent studies also propose different ways for automatic jailbreaking or red-teaming (Perez et al., 2022; Xu et al., 2024; Mazeika et al., 2024; Paulus et al.; Chao et al., 2025; Ge et al., 2024; Wang et al., 2025b).

**MCTS in LLMs.** Recent work has explored Monte Carlo Tree Search (MCTS) as a mechanism to enhance LLMs, primarily

for search-guided reasoning and planning. Methods such as MCTS-based planners apply tree search over intermediate reasoning steps to improve solution quality for individual inputs, often guided by prompted or learned value functions (Xie et al., 2024; Feng et al., 2023; Zhang et al., 2024a; Koh et al., 2024; Hu et al., 2025; Pouplin et al., 2024). A related line of work adopts AlphaZero-style frameworks, using MCTS as a policy improvement operator to collect high-quality trajectories for self-training or reinforcement learning (Zhang et al., 2024b; Feng et al., 2023; Wei et al., 2025). Tree search has also been successfully applied in domain-specific settings such as formal theorem proving, where external verifiers provide sparse but reliable feedback (Xin et al., 2024). In contrast to these approaches, which are largely solution-centric and aim to improve model accuracy, our work repurposes MCTS for adaptive evaluation, treating probing as a budgeted exploration problem over the failure space and using search to systematically discover recurring failure modes rather than optimal reasoning trajectories.

**Active Learning in LLMs.** Recent work has increasingly incorporated LLMs into active learning frameworks to reduce annotation cost and improve data efficiency. A common paradigm treats LLMs as supervision oracles, where informative samples selected by uncertainty or diversity criteria are labeled by LLMs instead of humans, covering applications such as cross-task text classification, generalized category discovery, and process reward model training (Zhang & Takada, 2025; Astorga et al., 2024; Duan et al., 2025; Liang et al., 2024). Beyond label acquisition, LLMs have also been used to guide data generation or augmentation within active learning loops, enabling the synthesis of challenging or underrepresented samples based on model feedback, including evolving knowledge distillation and safety-critical data construction (Liu et al., 2024; Hassan et al., 2024). Other studies explore LLM-driven querying strategies to alleviate cold-start or few-shot limitations, exemplified by ActiveLLM (Bayer et al., 2026), while recent work further extends active learning to cost-aware and partially observed settings using LLMs as generative surrogates for uncertainty estimation (Astorga et al., 2024). Comprehensive surveys systematize these directions by categorizing LLM-based active learning methods according to their roles in selection, generation, and annotation (Xia et al., 2025). In contrast to prior work that primarily applies active learning to improve downstream model performance or reduce labeling cost, our work adopts active learning as a mechanism to iteratively select informative interaction instances based on model feedback, enabling systematic exploration and analysis of LLM behaviors rather than model training.

**Comparison With Previous Work.** We compare PROBELLM with representative automated evaluation frameworks along several complementary dimensions, as summarized in Table 6. In particular, we focus on whether a method can identify structured failure modes rather than isolated error cases, ensure the truthfulness of generated test cases, operate without explicit human intervention, remain independent of fixed benchmarks, and provide coverage over diverse failure modes.

ACE (Afkanpour et al., 2025) emphasizes capability-level decomposition and efficient evaluation through active learning, enabling broad benchmark-agnostic assessment. However, its evaluation primarily reflects capability gaps rather than explicitly characterizing structured failure modes, and its diversity emerges implicitly from capability coverage. AutoDetect (Cheng et al., 2024) is designed to surface model failures through heuristic- or pattern-driven procedures, but it mainly operates at the level of individual failure cases and does not explicitly model or analyze higher-level failure modes or their diversity. AutoBencher (Li et al., 2025) focuses on benchmark construction and curation, offering stronger control over test data quality, but it typically requires substantial human intervention and remains tied to static benchmark settings. Moreover, most of these works do not introduce explicit data quality control when synthesizing test cases (e.g., directly using LLMs for question generation). Wang et al. (2025b) propose an automated method for generating distractor questions from original benchmark items to evaluate model robustness, but the generated cases remain closely coupled to existing benchmarks and do not induce structured failure modes.

More recent work such as FACT-AUDIT (Lin et al., 2025) adopts an adaptive, multi-agent framework to dynamically audit factual reasoning failures in LLMs. While it is benchmark-agnostic and effective at discovering diverse failure instances within fact-checking tasks, it does not explicitly consolidate failures into structured, interpretable failure modes. Similarly, APT (Rao et al., 2025) focuses on automatically acquiring weakness cases for iterative preference training, aiming to improve model performance rather than to analyze or characterize failure modes. As a result, these methods remain largely case-centric and do not provide mode-level diagnostics. Other related approaches, such as SEA (Song et al., 2025), focus on efficiently harvesting large numbers of errors in knowledge-base–driven settings. In contrast, PROBELLM is designed to automatically discover and induce structured failure modes via macro-level probing strategy search, providing explicit, interpretable explanations of how and why a model fails. Its fully automated evaluation pipeline requires no human intervention, while leveraging agentic tools to enforce data quality control by ensuring the truthfulness and validity of generated test cases.

# B. Details of PROBELLM

## B.1. PROBELLM Algorithm Process

The overall algorithm of PROBELLM is as follows:

**Initialization.** Initialize the failure pool $\mathcal{F}$ using a seed dataset $\mathcal{S}_0$ by probing until $n$ failure cases are obtained. Each failure initializes a seed node under both MACRO and MICRO search strategy roots, with node statistics initialized accordingly.

**Adaptive probing (§3.2 & §3.3).** Repeat hierarchical MCTS rollouts until a stopping criterion is met. In each rollout, select a search strategy (MACRO/MICRO) via UCB, traverse the corresponding tree via UCB selection, expand under the width constraint by sampling $x_{\text{new}} \sim G_{\text{s}}(\cdot \mid u)$, evaluate and verify $r_{\text{new}}$, and back up statistics along the selected path. Add $x_{\text{new}}$ to $\mathcal{F}$ if $r_{\text{new}} = 1$. The probing process stops when either the total number of simulations reaches a predefined budget or the number of discovered failure cases reaches a target threshold.

**Failure-mode construction (§3.4).** Compute failure-aware embeddings for cases in $\mathcal{F}$, cluster embeddings into $\{\mathcal{C}_k\}_{k=1}^{K}$, and induce each cluster with boundary-aware evidence selection to obtain failure-mode descriptions.

**Output.** Return the discovered failure modes and their associated failure cases for analysis and downstream mitigation.

## B.2. Scope of Test Cases

PROBELLM generates test cases with well-defined ground-truth answers. In our current instantiation, model outputs are evaluated using an LLM-based judge; however, the presence of ground truth provides a stable reference that substantially improves the reliability of these judgments. Compared to open-ended settings where correctness is inherently subjective, ground-truth-anchored questions constrain the judge's role to verifying factual or rule-based consistency, reducing ambiguity and variance in failure signals. The PROBELLM framework itself is not restricted to closed-form questions and can be extended to open-ended scenarios by adopting calibrated or judging mechanisms, which we leave for future work.

## B.3. Tool Details

**Python Execution.** The Python execution environment plays a dual role in PROBELLM. Beyond serving as a testing sandbox, it acts as a tool-augmentation layer that allows the generator to invoke Python during reasoning for exact computation (e.g., numerical calculation and data processing). This design instantiates a *tool-augmented reasoning* paradigm, reducing errors caused by heuristic or approximate reasoning.

**Perturbation.** Perturbation is used to probe intrinsic model brittleness rather than knowledge gaps. All perturbations are constrained to preserve the original semantics, ensuring that observed failures are attributable to model vulnerability under minimal input variation.

**Web Retrieval.** Web retrieval powered by OpenAI [4] supports both answer construction and self-verification. Retrieved evidence is used to validate factual correctness and reference consistency, providing an interpretable basis for downstream analysis.

*Table 7.* Intuition behind boundary-aware evidence selection for failure-mode induction. Different evidence types play complementary roles: central failures reveal the core error, while boundary failures and nearby non-failures expose where the failure region begins and ends.

| Evidence Type | Selection Criterion | Evidence Contribution | Bias if Omitted |
|---|---|---|---|
| **Central failures** | Typical failures near the cluster center | The core error pattern the model repeatedly makes | The failure mode becomes vague or internally inconsistent |
| **Boundary failures** | Failures near the edge of the cluster | Where the model *starts* to fail under small changes | The induction over-generalizes the failure scope |
| **Nearby non-failures** | Closest correct cases to boundary failures | What the model can still do *correctly* in similar contexts | The LLM for induction cannot tell what distinguishes failure from success |

---

[4]https://platform.openai.com/docs/guides/tools-web-search

## B.4. PROBELLM Algorithm

---

**Algorithm 1** PROBELLM

---

**Require:** Seed dataset $\mathcal{S}_0$, target model $f_\theta$, verifier $V$, simulation budget $T_{\max}$, failure budget $M$
**Ensure:** Failure modes $\{\mathcal{C}_k\}_{k=1}^K$ with summaries, and failure pool $\mathcal{F}$
 1: *// Stage 1: Initialization*
 2: $\mathcal{F} \leftarrow \emptyset$; initialize MACRO/MICRO roots
 3: Probe $\mathcal{S}_0$ to collect $n$ initial failure cases; add them to $\mathcal{F}$ as seed nodes
 4: *// Stage 2: Adaptive probing (MCTS)*
 5: $t \leftarrow 0$
 6: **while** $t < T_{\max}$ **and** $|\mathcal{F}| < M$ **do**
 7:     Select mode $\mathsf{s} \in \{\text{MACRO}, \text{MICRO}\}$ via UCB
 8:     Traverse within the mode tree via UCB to a node $u$ eligible for expansion
 9:     Sample and attach $x_{\text{new}} \sim G_{\mathsf{s}}(\cdot \mid u)$ (width-capped)
10:     Query $y \leftarrow f_\theta(x_{\text{new}})$; obtain $\mathbb{I}_{\text{fail}}(x) \triangleq \mathbb{1}\big[V\big(f_\theta(x), y^\star(x)\big) = 0\big]$
11:     **if** $\mathbb{I}_{\text{fail}}(x) = 1$ **then**
12:         $\mathcal{F} \leftarrow \mathcal{F} \cup \{x_{\text{new}}\}$
13:     **end if**
14:     Backpropagate visit and failure statistics along the selected path
15:     $t \leftarrow t + 1$
16: **end while**
17: *// Stage 3: Mode extraction (embed & cluster)*
18: Compute failure-aware embeddings $\{z(x)\}_{x \in \mathcal{F}}$ and cluster into $\{\mathcal{C}_k\}_{k=1}^K$
19: *// Stage 4: Boundary-aware induction*
20: **for** $k = 1, 2, \ldots, K$ **do**
21:     **if** $|\mathcal{C}_k| < \tau$ **then**
22:         Prompt LLM for induction with all failures in $\mathcal{C}_k$; obtain summary $s_k$
23:     **else**
24:         Compute centroid $\mu_k$ of $\mathcal{C}_k$ in embedding space
25:         Select $n_c$ central failures $\mathcal{S}_k^{\text{core}} \subset \mathcal{C}_k$ (closest to $\mu_k$)
26:         Select $b$ boundary failures $\mathcal{S}_k^{\text{bd}} \subset \mathcal{C}_k$ (most peripheral/low-density)
27:         **for** each $x \in \mathcal{S}_k^{\text{bd}}$ **do**
28:             Retrieve nearest verified non-failure $x^+$ from the search tree (by embedding distance)
29:         **end for**
30:         Prompt LLM for induction with $\mathcal{S}_k^{\text{core}}$ and pairs $\{(x, x^+) : x \in \mathcal{S}_k^{\text{bd}}\}$; obtain summary $s_k$
31:     **end if**
32: **end for**
33: *// Output*
34: **Output:** $\{(\mathcal{C}_k, s_k)\}_{k=1}^K$ and $\mathcal{F}$

---

## C. Experiment Details

### C.1. Setup Details

**Model details.** We show the details of selected models used in the experiments in Table 8.

*Table 8.* Target model list in the experiments with vendor and whether weights are publicly downloadable (open-weight).

| Model | Vendor | Open-weight? |
|---|---|---|
| Claude-3.5-sonnet (Anthropic, 2024) | Anthropic | ✗ |
| Deepseek-v3.2 (DeepSeek-AI, 2025) | DeepSeek | ✓ |
| devstral-2512 (Mistral AI, 2025a) | Mistral AI | ✓ |
| ministral-8b-2512 (Mistral AI, 2025b) | Mistral AI | ✓ |
| Gemini-2.5-flash (Google, 2025) | Google | ✗ |
| GPT-4o-mini (OpenAI, 2024) | OpenAI | ✗ |
| GPT-oss-20b (OpenAI, 2025) | OpenAI | ✓ |
| Llama-3.1-8b-instruct (Meta, 2024) | Meta | ✓ |
| phi-4 (Abdin et al., 2024) | Microsoft | ✓ |
| Grok-4.1-fast (xAI, 2025) | xAI | ✗ |
| olmo-3-7b-instruct (Olmo et al., 2025) | AI2 | ✓ |
| granite-4.0 (Soule & Bergmann, 2025) | IBM | ✓ |

**Benchmark details**. We select five widely-used benchmarks in the experiments:

- **MMLU (Measuring Massive Multitask Language Understanding)** (Hendrycks et al., 2020): A multiple-choice benchmark covering dozens of academic disciplines, designed to evaluate a model's broad factual knowledge and reasoning ability across diverse domains.
- **SuperGLUE** (Wang et al., 2019): A collection of challenging natural language understanding tasks that test advanced reasoning, inference, and language comprehension beyond the original GLUE benchmark.
- **MBPP (Mostly Basic Python Problems)** (Austin et al., 2021): A code generation benchmark consisting of short Python programming tasks with natural language descriptions and unit tests, aimed at assessing basic coding proficiency.
- **HellaSwag** (Zellers et al., 2019): A commonsense reasoning benchmark where models must select the most plausible continuation of a given situation from multiple candidate endings.
- **TruthfulQA** (Lin et al., 2021): A benchmark that evaluates whether language models produce truthful answers and avoid common misconceptions, especially in domains where humans often answer incorrectly.

**Metric details.** We show three metrics' definition as follows:

- **Error Rate.** The error rate is defined as the proportion of generated questions for which the target model produces an incorrect answer, measured over all generated questions.
- **Noise Rate (NR).** The noise rate is defined as the proportion of discovered test cases that are identified as noise by HDBSCAN and thus not assigned to any failure-mode cluster.
- **Cluster-Size Standard Deviation (Std$_{\text{CS}}$).** Let $\{|\mathcal{C}_k|\}_{k=1}^{K}$ denote the sizes of the discovered failure-mode clusters. Std$_{\text{CS}}$ is computed as the standard deviation of $\{|\mathcal{C}_k|\}_{k=1}^{K}$.

**Other details.** For each benchmark, different runs are initialized with identical seeds, ensuring that the same initial samples are used and that the experimental setup is fully reproducible. We control the probing budget by fixing the number of simulations to 100 per benchmark. Consequently, each target model is evaluated with a total of 500 simulations across the five benchmarks used in our experiments.

Regarding tool usage, we apply special handling for Python execution failures. When a python_exec call fails (e.g., due to runtime errors or invalid code), the error message is fed back to the generator model, which is allowed to retry up to three times. If all retry attempts fail, the corresponding generation is discarded and does not contribute to the probing budget.

*Table 9.* Human evaluation results. Values are reported as $\mu_\sigma$ (mean with standard deviation in the subscript).

| Eval | Metric | Score ($\mu_\sigma$) |
|---|---|---|
| **E1** | Answerable | $0.936_{0.076}$ |
| | Unambiguous | $0.964_{0.059}$ |
| | Correctness | $0.936_{0.100}$ |
| **E2** | Correctness | $1.000_{0.000}$ |
| **E3** | Accuracy | $0.933_{0.091}$ |
| **E4** | Faithfulness | $4.268_{0.599}$ |
| | Specificity | $4.368_{0.418}$ |
| | Usefulness | $4.468_{0.364}$ |

*Figure 10.* Human evaluation of question quality (E1) across different frameworks, measuring answerability, unambiguity, and correctness.

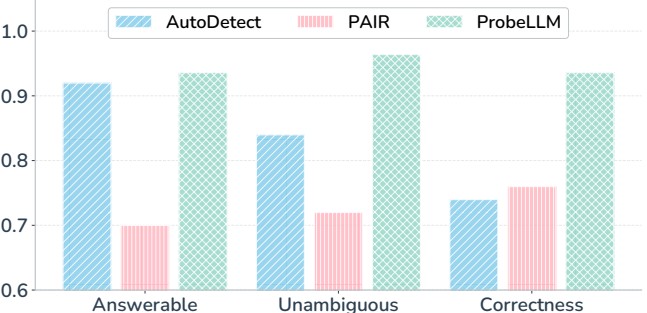

*Table 10.* The results of another two generator: `Claude-4.5-sonnet` and `Llama-3.3-70b-instruct` across 4 target models. Avg. ER means average error rate and cluster means the number of clusters.

| Target Model | MMLU Sim. Type MA. | MI. | Error Rate(%) | SuperBLUE Sim. Type MA. | MI. | Error Rate(%) | MBPP Sim. Type MA. | MI. | Error Rate(%) | TruthfulQA Sim. Type MA. | MI. | Error Rate(%) | Avg ER | # Cluter |
|---|---|---|---|---|---|---|---|---|---|---|---|---|---|---|
| **Claude-4.5-sonnet** | | | | | | | | | | | | | | |
| Deepseek-v3.2 | 33% | 67% | 37% | 23% | 77% | 41% | 22% | 78% | 35% | 19% | 81% | 50% | 41% | 9 |
| Gemini-2.5-flash | 38% | 62% | 28% | 21% | 79% | 45% | 50% | 50% | 20% | 33% | 67% | 31% | 31% | 6 |
| GPT-5.2 | 34% | 66% | 26% | 31% | 69% | 31% | 21% | 79% | 43% | 31% | 69% | 28% | 32% | 10 |
| ministral-8b | 37% | 63% | 56% | 21% | 79% | 64% | 41% | 59% | 46% | 32% | 68% | 53% | 55% | 13 |
| **Llama-3.3-70b-instruct** | | | | | | | | | | | | | | |
| Deepseek-v3.2 | 25% | 75% | 63% | 40% | 60% | 42% | 12% | 88% | 70% | 32% | 68% | 48% | 56% | 12 |
| Gemini-2.5-flash | 19% | 81% | 67% | 41% | 59% | 39% | 18% | 82% | 71% | 29% | 71% | 47% | 56% | 11 |
| GPT-5.2 | 35% | 65% | 34% | 43% | 57% | 35% | 29% | 71% | 54% | 24% | 76% | 59% | 46% | 10 |
| ministral-8b | 37% | 63% | 58% | 34% | 66% | 56% | 32% | 68% | 75% | 30% | 70% | 56% | 61% | 14 |

## C.2. Baseline Comparison

Figure 10 reports the human evaluation of question quality (E1) across different frameworks. PROBELLM consistently achieves the highest scores on answerability, unambiguity, and correctness, indicating that it generates higher-quality and more reliable test questions. This improvement can be attributed to PROBELLM's use of agentic tools such as web search and Python execution, which enable evidence-grounded question construction and precise constraint verification.

## C.3. Human Evaluation

Figure 9 reports consistent human-evaluation results across all protocols. **E1** confirms that synthesized test cases are largely answerable, unambiguous, and paired with correct ground truths, validating data quality. **E2** shows perfect agreement between the LLM-based verifier and human judgments, supporting the reliability of automatic verification. **E3** indicates that discovered clusters are coherent and well separated, while **E4** demonstrates that the induced failure-mode descriptions are faithful, specific, and diagnostically useful.

## C.4. Analysis of Different Generators

Table 10 reports results using `Claude-4.5-sonnet` and `Llama-3.3-70b-instruct` as alternative generators. Across both generators and all four target models, PROBELLM consistently induces substantial error rates and discovers multiple failure clusters, demonstrating that its effectiveness is not tied to a specific generator. While `Llama-3.3-70b-instruct` generally yields higher average error rates and more clusters than `Claude-4.5-sonnet`, the relative trends across target models remain consistent, indicating that PROBELLM is robust to generator choice and scales with stronger generators to expose richer and more severe failure modes.

## C.5. MACRO–MICRO Allocation Analysis

We analyze the relationship between the MACRO/MICRO simulation allocation and the resulting error rate across all target models and benchmarks. As shown in the Figure 11, we do not observe a monotonic or linear correlation between the Mi/Ma ratio and the error rate. High error rates consistently arise under a wide range of Macro–Micro proportions, including both Micro-heavy and more balanced regimes. This suggests that failure discovery effectiveness in PROBELLM is not driven by a fixed search ratio. Instead, the results highlight the importance of adaptive budget allocation: MACRO exploration is necessary to surface diverse failure regions, while MICRO refinement consolidates evidence within promising areas. Over-emphasizing either regime alone does not reliably improve failure discovery, reinforcing the design choice of treating MACRO and MICRO as complementary, dynamically selected components rather than tunable hyperparameters.

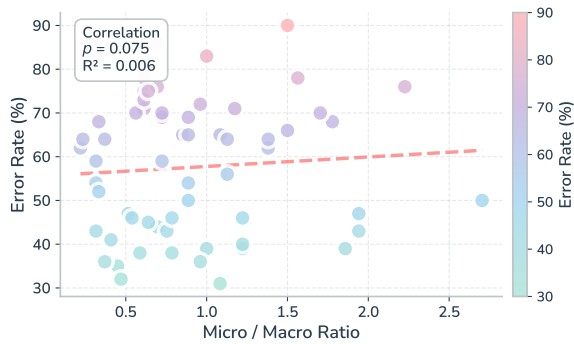

*Figure 11.* Relationship between the MACRO/MICRO simulation ratio and the error rate across all models and benchmarks.

## C.6. Mitigation Experiment

**Experimental setup.** For training, we construct error-focused datasets by mixing PROBELLM-discovered incorrect samples with correctly answered ones, where the total training size is varied to study the effect of data scale; this mixture is used to prevent forgetting on previously correct behaviors (Huang et al., 2025c). For evaluation, we consider two sets: (i) an error-prone set consisting of high–error-rate samples discovered by PROBELLM across five datasets, where most instances are answered incorrectly by the base model; and (ii) a general evaluation set randomly sampled from the remaining portions of the existing benchmarks to assess capability preservation. Training is performed using GRPO (Shao et al., 2024) with an LLM-as-a-Judge (GPT-5.2) verifier, assigning a reward of 1 for correct answers and 0 otherwise, with 8 rollouts per instance and 2 training epochs.

## C.7. Model Evolution Analysis

In this section, we provide a granular analysis of how model failure modes evolve across generations, combining the quantitative metrics from Table 11 and the topological visualization in Figure 7.

The quantitative trajectory reveals a significant improvement in robustness. For the OpenAI family, we observe a sharp decline in both the Average Error Rate (from 75% in GPT-3.5 to 29% in GPT-5.2) and the number of identified failure clusters (from 16 down to 5). This reduction suggests that newer models are not only becoming more accurate but are also closing off entire categories of logical loopholes. A similar trend is observed in the Mistral family, where the error rate drops from 81% to 57%, indicating that open-weights models are following a parallel evolutionary path, albeit with a lag in robustness convergence compared to closed-source counterparts.

The structural clarity of the UMAP visualization in Figure 7 further attests to PROBELLM's capability to disentangle and map the complex landscape of model deficits. By projecting the failure queries into a semantic space, PROBELLM effectively distinguishes between different categories of weaknesses, revealing a nuanced evolutionary relationship. The visualization clearly separates transient failure modes, which are represented by isolated dark blue centroids surrounded by older model nodes, from persistent logical flaws indicated by connected purple centroids. This topological distinction demonstrates that PROBELLM does not simply aggregate errors, but successfully identifies the lineage of failure modes, allowing us to trace which weaknesses are resolved across versions and which remain stubborn, structural deficits.

Furthermore, PROBELLM proves effective in pinpointing the shifting domain specificity of these weaknesses. As highlighted by the distinct insets, the framework detects that failure modes in advanced models retreat into increasingly niche scientific corners. For instance, PROBELLM identifies a distinct failure cluster for GPT-5.2 centered on the highly specialized domain of *"Hyperfine Splitting in EPR Spectroscopy"*. This ability to isolate distinct, domain-specific clusters confirms that PROBELLM can adaptively probe the deepening knowledge boundaries of evolving models, exposing that as models scale, their vulnerability surface morphs from general complexity to extreme disciplinary specialization.

*Table 11.* Main results across 8 target models (OpenAI models are shown in light blue and Mistral-family models in light purple). We report MACRO (MA.), MICRO (MI.), and the error rate (%).

| Target Model | MMLU Sim. Type | | Error | SuperGLUE Sim. Type | | Error | MBPP Sim. Type | | Error | TruthfulQA Sim. Type | | Error | Avg. ER | # Cluster |
|---|---|---|---|---|---|---|---|---|---|---|---|---|---|---|
| | MA. | MI. | Rate$_{(\%)}$ | MA. | MI. | Rate$_{(\%)}$ | MA. | MI. | Rate$_{(\%)}$ | MA. | MI. | Rate$_{(\%)}$ | | |
| **OpenAI Family** | | | | | | | | | | | | | | |
| GPT-3.5-turbo (Mar 2023) | 26% | 74% | 76% | 16% | 84% | 78% | 26% | 74% | 80% | 22% | 78% | 67% | 75% | 16 |
| GPT-4.1 (Apr 2025) | 34% | 66% | 31% | 22% | 78% | 42% | 50% | 50% | 22% | 23% | 77% | 47% | 36% | 6 |
| GPT-4o-mini (Jul 2024) | 16% | 84% | 71% | 18% | 83% | 57% | 20% | 80% | 65% | 29% | 71% | 57% | 63% | 10 |
| GPT-5.2 (Dec 2025) | 32% | 68% | 27% | 36% | 64% | 21% | 35% | 65% | 25% | 23% | 77% | 41% | 29% | 5 |
| **Mistral Family** | | | | | | | | | | | | | | |
| Mistral-7B-Ins.  (May 2024) | 19% | 81% | 81% | 27% | 73% | 65% | 13% | 87% | 87% | 38% | 62% | 77% | 78% | 20 |
| Mistral-7B-Ins.-v0.1 (Sep 2023) | 23% | 77% | 86% | 19% | 81% | 72% | 31% | 69% | 82% | 21% | 79% | 82% | 81% | 16 |
| Ministral-8B (Oct 2024) | 27% | 73% | 81% | 14% | 86% | 67% | 32% | 68% | 77% | 24% | 76% | 72% | 74% | 19 |
| Ministral-3-8B-2512 (Dec 2025) | 14% | 86% | 59% | 21% | 79% | 60% | 31% | 69% | 49% | 22% | 78% | 60% | 57% | 16 |

## C.8. Clustering Sensitivity Analysis

For sensitivity analysis, we vary the key HDBSCAN hyperparameters that control clustering granularity. Specifically, we sweep `min_cluster_size` over $\{3, 6, 9, 12, 15\}$ and `min_samples` over $\{1, 3, 5, 7, 9\}$, resulting in a diverse set of clustering configurations spanning different density and stability regimes.

To assess the robustness of failure-mode discovery to clustering hyperparameters, we measure *conditional pairwise co-assignment consistency* across multiple HDBSCAN configurations. Let $\mathcal{F} = \{u_1, \ldots, u_N\}$ denote the set of discovered failure cases and $\mathcal{S} = \{s_1, \ldots, s_M\}$ denote different clustering settings. For a configuration $s \in \mathcal{S}$, each failure case is assigned a cluster label $c^{(s)}(u) \in \{1, \ldots, K_s\} \cup \{\text{NOISE}\}$.

For a pair of failure cases $(u_i, u_j)$, we define a co-assignment indicator under $s$ as

$$\mathbf{1}_{ij}^{(s)} = \mathbb{I}\big[c^{(s)}(u_i) = c^{(s)}(u_j) \neq \text{NOISE}\big].$$

The cross-configuration consistency of the pair is

$$\text{Cons}(u_i, u_j) = \frac{1}{|\mathcal{S}|} \sum_{s \in \mathcal{S}} \mathbf{1}_{ij}^{(s)}.$$

Since most failure pairs are unrelated, we report a *conditional* consistency score by averaging $\text{Cons}(u_i, u_j)$ over pairs that co-occur in the same non-noise cluster under at least one configuration.

Figure 12 reports the conditional pairwise co-assignment consistency across four target models. All models exhibit consistently high consistency scores, with mean values between $0.55$ and $0.82$, indicating that discovered failure modes are largely stable under different clustering hyperparameters. The median consistency is high across models, suggesting that the core structure of dominant failure modes remains invariant despite changes in clustering granularity. Overall, these results demonstrate that PROBELLM's failure-mode discovery is not sensitive to specific clustering parameter choices.

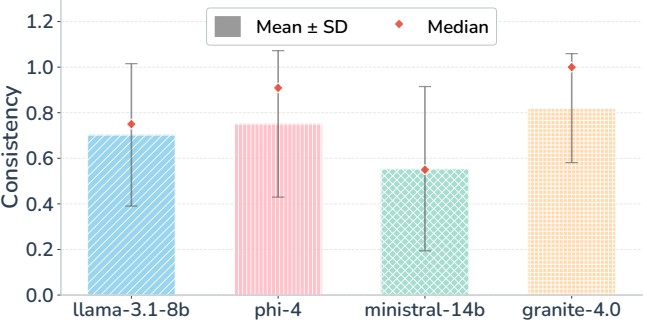

*Figure 12.* Clustering sensitivity analysis. Conditional pairwise co-assignment consistency across different clustering settings.

## C.9. Cluster Overview

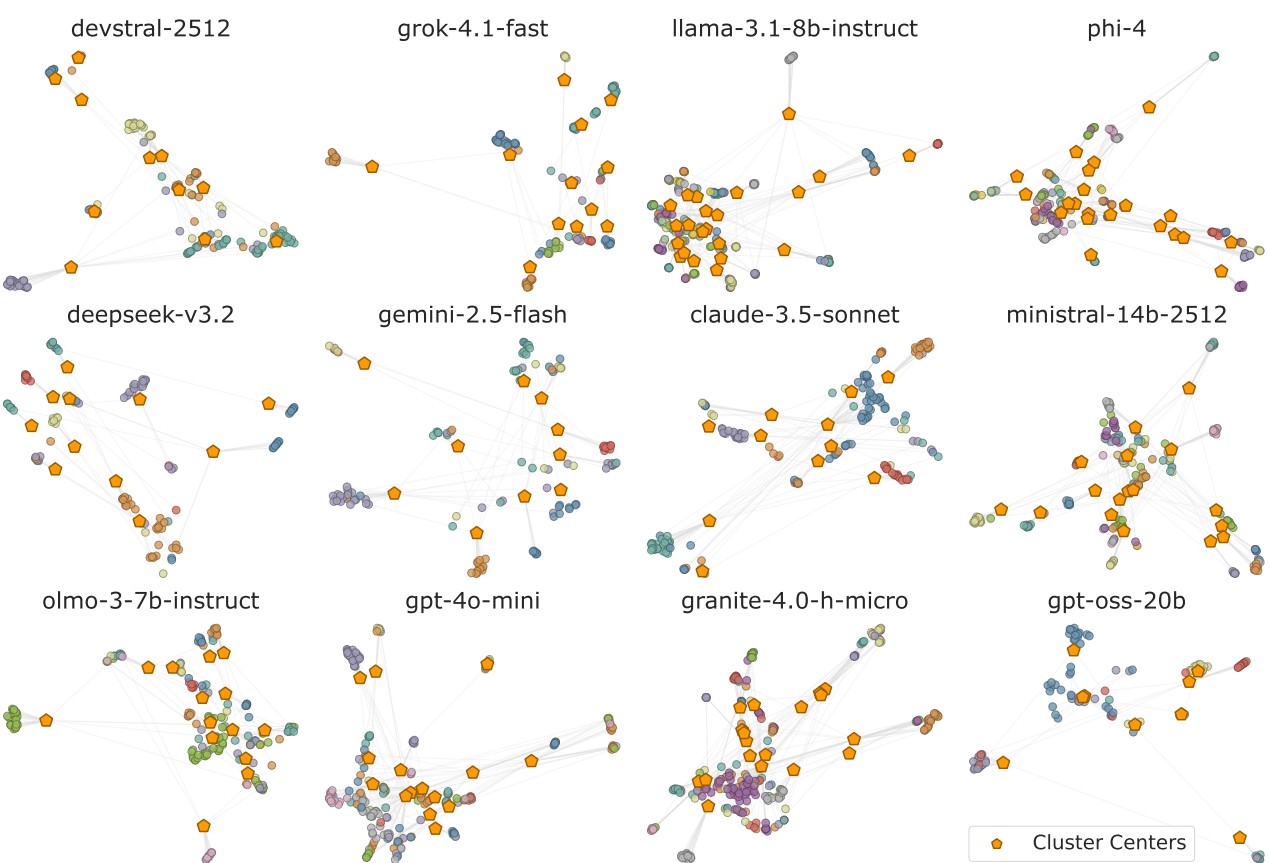

*Figure 13.* UMAP visualization of vulnerability embeddings identified by PROBELLM across 12 LLMs. Small nodes represent individual failure samples, while orange pentagons indicate the centroids of specific vulnerability topics. Faint lines connect samples to their respective centers.

Figure 13 presents a comparative UMAP visualization of the vulnerability embeddings identified by PROBELLM across twelve different LLMs. By examining the topological structure and the distribution of cluster centroids across these models, we identify two distinct patterns that reveal the underlying nature of their safety deficits: one characterized by structural consolidation and the other by fragmentation.

The first pattern, exemplified by highly capable models such as `Gemini-2.5-Flash` and `Claude-3.5-Sonnet`, manifests as a consolidated topology characterized by a low number of distinct cluster centroids. In these visualizations, the failure samples aggregate into a few well-defined, compact groups centered around a limited set of orange pentagons. This structural simplicity suggests that the safety deficits of these advanced models are not random but are concentrated in specific, high-level semantic categories. The clear separation between these few clusters indicates that the models maintain robust performance across most domains, failing only when triggered by highly consistent and identifiable vulnerability patterns. In contrast, the second pattern is evident in models like `Llama-3.1-8b-Instruct`, which exhibit a highly fragmented topology populated by a large number of cluster centroids. The visualization displays a complex archipelago of numerous, smaller clusters rather than a few dominant ones. This high cardinality of centroids indicates a diverse and widespread range of failure modes, suggesting that the model's weaknesses are not confined to a few specific topics but are scattered across a broader spectrum of semantic variations. Such a fragmented structure implies a lower degree of generalization in safety alignment, where the model exhibits fragility across many distinct, fine-grained scenarios rather than converging into a few systematic behaviors.

# D. Baseline Details

**Disclaimer.** We note that these methods are designed with different objectives and evaluation targets (e.g., PAIR optimizes attack success, not failure-mode coverage; we include it as a reference point rather than a strict competitor); accordingly, our comparison is intended solely to provide a reference under comparable settings, rather than to diminish or subsume the contributions of these baselines.

**PAIR (Chao et al., 2025).** We implement PAIR as an iterative adversarial prompting framework following its original design. For each run, we initialize PAIR with questions sampled from five benchmarks: MMLU, SuperGLUE, MBPP, HellaSwag, and TruthfulQA, drawing 10 questions from each benchmark. Each question is first issued to the *Target* model to obtain a response. A *Judge* model then evaluates each question–answer pair and assigns a score in $[1, 10]$, where lower scores indicate poorer performance. We set the threshold to $k = 5$. If the score is below $k$, the question is treated as a low-score case and passed to the *Questioner*, which generates a new adversarial question conditioned on the tuple (previous question, target response, judge score). This refinement process continues until either the target model achieves a score $\geq k$ or a maximum of three refinement steps is reached. All generated questions, responses, and scores are recorded. We run PAIR for 200 independent trials under this setting.

**AutoDetect (Cheng et al., 2024).** We implement AutoDetect as a multi-stage weakness discovery and targeted probing framework. For each run, we sample 5 initial questions from each of the same five benchmarks used in PAIR. These questions are directly evaluated by the *Target* model, and a *Judge* assigns integer scores in $[1, 5]$ using ground-truth references, with lower scores indicating more severe failures. We set the threshold to $k = 3$, so scores in $\{1, 2\}$ are considered low-score cases. In each iteration, if more than two low-score cases are identified, they are aggregated by an *Assessor* to induce high-level evaluation topics representing recurring failure patterns. The number of induced topics is flexible, ranging from one up to the total count of low-score cases. Subsequently, the *Questioner* generates three specifically designed, more challenging questions for *each* induced topic by increasing constraints or reasoning complexity. The entire process runs for at most three rounds (including the initial benchmark round), and terminates early if no new low-score topics are identified. AutoDetect is also executed for 200 independent trials, with all intermediate results logged.

# E. Theoretical Proof

## E.1. Feasibility of MACRO-MICRO Probing

In this part, we aim to give the proof of the feasibility of solidly identifying a failure mode from a single failure case via MICRO case provided by MACRO selection.

To formalize our proof, we need to first define the semantic space and the behavior of the model failures within that space. Let $\mathcal{X} \subseteq \mathbb{R}^d$ be a high-dimensional semantic space. We define a failure function $f : \mathcal{X} \to \{0, 1\}$, where $f(x) = 1$ indicates a model failure. The failure probability field is defined as $P(x) = \Pr(f(x) = 1)$.

**Definition E.1** (Failure Mode). A failure mode $\mathcal{M} \subset \mathcal{X}$ is a contiguous sub-region in the semantic space where the average failure probability exceeds a significant threshold $\tau$, i.e., $\mathbb{E}_{x \sim \mathcal{M}}[P(x)] \geq \tau$.

**Assumption E.2** (Local Semantic Coherence). LLM failures are not isolated stochastic events but result from systematic knowledge gaps. Based on prior findings in behavioral testing and systematic error discovery that model failures often cluster over semantically similar perturbations and contiguous regions in representation space (Ribeiro et al., 2020; d'Eon et al., 2021). Thus, we can begin with a benign assumption or description, if a failure case $x_0$ is identified ($f(x_0) = 1$), there exists a local neighborhood $B(x_0, \epsilon) \subseteq \mathcal{M}$ such that for any $x \in B(x_0, \epsilon)$, $P(x) \geq p_{min} > 0$.

**Theorem E.3** (Mode Identification Convergence). *Given an initial failure $x_0$, the MICRO probing strategy, which samples $n$ independent semantic variants in $B(x_0, \epsilon)$, identifies the failure mode with probability $P_{succ}$ satisfying:*

$$P_{succ} \geq 1 - (1 - p_{min})^n \tag{12}$$

*As $n \to \infty$, $P_{succ}$ converges to 1 exponentially.*

*Proof.* Let $E_i$ be the event that the $i$-th MICRO probe fails to identify a failure. By the assumption of local coherence, $\Pr(E_i) \leq 1 - p_{min}$. Since MICRO probes are generated through independent semantic perturbations, the joint probability that all $n$ probes fail to surface a failure is $\Pr(\cap_{i=1}^{n} E_i) \leq (1 - p_{min})^n$. The probability of identifying at least one additional failure (and thus confirming the mode) is $1 - \Pr(\cap_{i=1}^{n} E_i)$. This completes the proof. $\square$

## E.2. Regret Bound of PROBELLM Framework

In this section, we analyze the search efficiency of PROBELLM by modeling its hierarchical MACRO-level selection as a tree-structured multi-armed bandit problem. Our goal is to prove that the cumulative regret grows sub-linearly, ensuring that the framework asymptotically converges to the most effective failure discovery strategy.

**Definition E.4** (Cumulative Regret). Let $T$ be the total budget of probes. For each time step $t \in [1, T]$, let $i_t$ be the thematic area selected by the MCTS policy, and $\mu_{i_t}$ be the expected failure discovery rate of that area. Let $\mu^*$ be the optimal discovery rate in the entire search space. The cumulative regret $R_T$ is defined as:

$$R_T = T\mu^* - \sum_{t=1}^{T} \mathbb{E}[\mu_{i_t}] \tag{13}$$

**Lemma E.5** (Node-Level Regret Bound). *Following the finite-time analysis of the UCB1 algorithm (Auer et al., 2002), for any node $v$ in the MACRO tree with $K$ children, the expected number of times a sub-optimal child $j$ (where $\Delta_j = \mu^* - \mu_j > 0$) is selected after $n$ visits to $v$ is bounded by:*

$$\mathbb{E}[N_j(n)] \leq \frac{8 \ln n}{\Delta_j^2} + \left(1 + \frac{\pi^2}{3}\right) \tag{14}$$

**Theorem E.6** (Search Convergence). *Under the MCTS-UCB selection policy, the average regret $\frac{R_T}{T}$ converges to zero as $T \to \infty$. That is, the framework's selection strategy is asymptotically optimal:*

$$\lim_{T \to \infty} \frac{R_T}{T} = 0 \tag{15}$$

*Proof.* The proof follows from the extension of UCB1 to tree structures, as established in the UCT algorithm (Kocsis & Szepesvari, 2006). First, by Lemma E.5, the regret at any individual decision node in the MACRO layer is $O(\ln T)$, which is sub-linear.

Second, Kocsis & Szepesvari (2006) proved that in a tree of finite depth $D$, the probability of selecting the optimal leaf (the most significant failure mode) converges to 1 as the number of samples $T$ increases. However, tree search can be sensitive to the "depth-first" bias. As noted by Coquelin & Munos (2007), while the worst-case regret in MCTS can be problematic, the introduction of the MICRO-probing stage in PROBELLM serves as a variance reduction mechanism. By providing more stable and truthful empirical estimates $\hat{\mu}_{i,t}$ through agentic verification, we satisfy the conditions for logarithmic regret bounds. Summing the regret across the tree, we obtain $R_T = O(\text{poly}(D) \cdot \ln T)$. Since $\lim_{T \to \infty} \frac{\ln T}{T} = 0$, the framework's discovery rate $\frac{1}{T} \sum \mu_{i_t}$ converges to the optimal rate $\mu^*$. $\qquad\square$

### E.3. Analysis of Time Complexity

**Notation Recap.** Throughout this analysis, we treat the cost of a single LLM query, embedding computation, and induction call as constant, and focus on the number of such calls and the remaining algorithmic overhead. Following prior notation in main text, $T$ denotes the maximum number of MCTS probing simulations, $D$ and $W$ the maximum search depth and branching factor, $M$ the failure pool size, $K$ the number of failure clusters, and $d$ the embedding dimension.

**Stage 1: MCTS-based Probing.** The probing stage performs at most $T$ Monte Carlo Tree Search simulations. Each simulation expands at most one new node and requires a constant number of LLM queries for test generation, execution, and verification. Let $C_{\text{query}}$ denote the cost of a single LLM query. Accordingly, the total cost of LLM queries in this stage scales linearly with the simulation budget, i.e., $O(TC_{\text{query}})$.

In addition to LLM calls, each simulation traverses the search tree to a bounded depth $D$ and selects among at most $W$ child nodes at each level. The tree traversal and update cost of a single simulation is $O(DW)$, yielding a total non-LLM computational cost of $O(TDW)$ for the probing stage.

**Stage 2: Failure Embedding and Clustering.** During probing, at most $M$ failure cases are collected. Each failure case is embedded once. Let $C_{\text{embed}}$ denote the cost of a single embedding computation. The total embedding cost is thus $O(MC_{\text{embed}})$.

The resulting $M$ failure embeddings are subsequently partitioned into $K$ clusters. Using a standard clustering algorithm with a bounded number of iterations (e.g., $k$-means), the clustering cost scales linearly with the number of data points, the number of clusters, and the embedding dimension. Consequently, the total clustering cost of this stage is $O(MKd)$.

**Stage 3: Boundary-Aware Collection and Induction.** In the final stage, each failure cluster is processed to extract representative and boundary cases. Computing distances between failure embeddings and their corresponding cluster centers requires a single pass over all failure cases, resulting in a cost of $O(Md)$.

For each selected boundary case, a corresponding non-failure reference is identified among previously explored samples. Since the number of explored samples is bounded by the probing budget, this step incurs an additional cost of $O(T)$.

Finally, one induction step is performed per cluster to summarize the corresponding failure mode. Let $C_{\text{induct}}$ denote the cost of a single induction call. The total induction cost of this stage is therefore $O(KC_{\text{induct}})$. Combining the above components, the overall cost of the third stage is $O(Md + T + KC_{\text{induct}})$.

**Overall Complexity.** Combining all three stages, the overall time complexity of the proposed method is $O(TDW + TC_{\text{query}}) + O(MC_{\text{embed}} + MKd) + O(Md + T + KC_{\text{induct}})$. In typical settings where $D$, $W$, $d$, and $K$ are small constants, the overall computational cost scales linearly with the probing budget $T$ and the number of collected failure cases $M$.

## F. Human Evaluation

### F.1. Evaluation Protocols

We complement automatic verification with the following human-evaluation protocols to validate (i) the quality of synthesized test cases and ground-truths, (ii) the correctness of failure evidence, and (iii) the reliability of extracted failure modes. Our framework assumes a verified reference answer $y^\star(x)$ and a verifier $V(\cdot, \cdot)$ for each test case $x$ (with model output $y = f_\theta(x)$), and it operationalizes *failure modes* as clusters of recurring failures in an embedding space. All protocols are implemented as lightweight web forms and are applied to stratified samples from both MACRO and MICRO probing regimes.

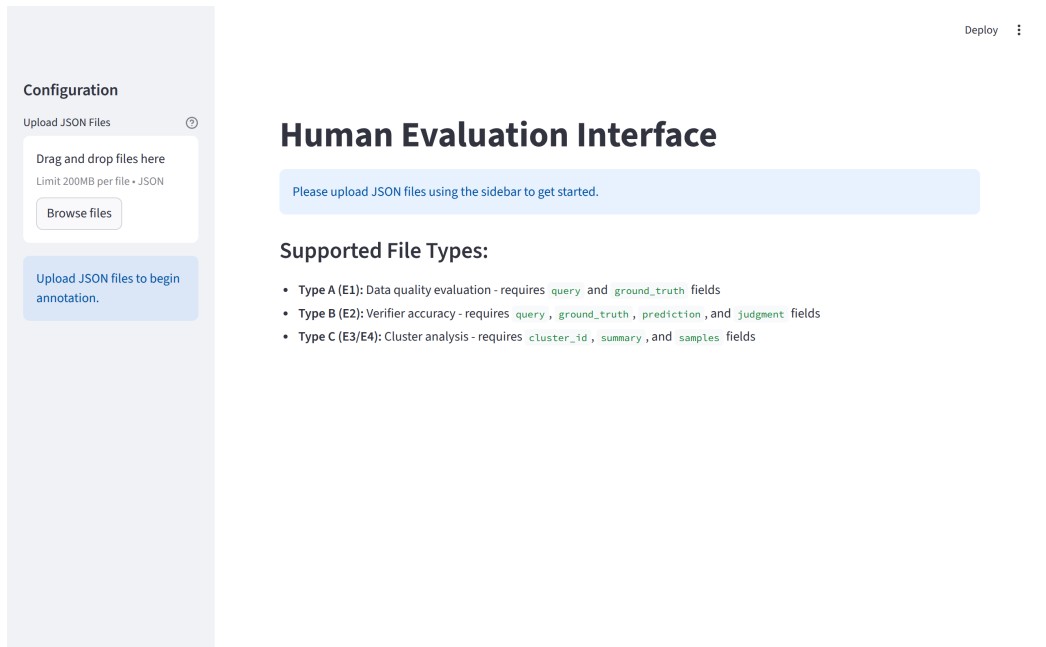

*Figure 14.* Human evaluation interface (1).

**E1: Test-case & ground-truth quality (Question Correctness).** Because our pipeline synthesizes questions and corresponding references, we must ensure that generated items are answerable, unambiguous, and paired with correct ground-truths. This directly audits the data-quality control goal of tool-augmented generation and verification.

Annotators are shown $(x, y^\star(x))$ together with the evidence used to support $y^\star(x)$. They label: (i) *Answerable* (Yes/No/Unsure), (ii) *Unambiguous* (Yes/No/Unsure), (iii) *Ground-truth correct* (Yes/No/Unsure), and (iv) a 5-point Likert score for relevance to systematic probing. When marking any item as "No", annotators provide a short rationale (e.g., missing constraints, multiple valid answers, evidence mismatch). We report acceptance rates and inter-annotator agreement.

**E2: Verifier accuracy (LLM Judge).** We rely on an LLM-based verifier/judge to determine whether the model output $y$ matches the reference $y^\star(x)$. To validate the reliability of these automatic judgments, we sample labeled instances and ask annotators to review $x$, $y$, and $y^\star(x)$ (with minimal supporting evidence when necessary) and provide a binary label indicating whether $y$ is *Correct* or *Incorrect*. We then compare the human label with the verifier decision (e.g., whether it predicts $V(y, y^\star) = 0$) and report the verifier accuracy, along with false-positive and false-negative rates when applicable.

**E3: Cluster coherence (Intruder Test).** A core claim of the method is that it discovers *structured failure modes* rather than isolated errors by clustering failures in representation space. We therefore test whether each cluster is internally coherent and separable from other clusters.

For each evaluated cluster, annotators see 5 representative failure cases: 4 sampled from the target cluster and 1 *intruder* sampled from a different cluster (matched by topic difficulty when possible). Annotators choose the intruder (A–E), rate cluster coherence on a 1–5 Likert scale, and optionally describe the common mechanism shared by the non-intruder items. We report intruder-identification accuracy and coherence scores.

**E4: Failure-mode description quality (induction Correctness).** After clustering, the system summarizes each cluster into

a natural-language failure-mode description. We evaluate whether these descriptions are faithful to the underlying evidence and specific enough to be diagnostically useful. This corresponds to the planned induction correctness audit.

Annotators are shown (i) several representative failures from one cluster and (ii) a candidate failure-mode description produced by the LLM for induction (optionally alongside a simple baseline description). They rate the candidate on three 1–5 Likert dimensions: *Faithfulness* (supported by the cluster evidence), *Specificity* (mechanism-level rather than generic), and *Diagnostic usefulness* (actionable for analysis/mitigation). When a baseline is shown, annotators also provide an overall preference (A/B/Tie) and optional edits to improve the candidate.

### F.2. Human Evaluation Metrics

We evaluate the proposed method using four human evaluation metrics. **E1**, **E2**, and **E4** are reported as *human agreement rates*, measuring the consistency between human judgments and the intended labels or model-generated outputs. Specifically, E1 measures agreement on test-case and ground-truth quality, E2 measures agreement between human judgments and the LLM-based verifier (i.e., LLM-as-a-Judge), and E4 measures agreement on the correctness and usefulness of induced failure-mode descriptions.

**E3** is reported as an *accuracy* metric, where annotators identify an intruder instance in each cluster. Higher intruder-identification accuracy indicates stronger cluster coherence and better separation between discovered failure modes.

### F.3. Human Evaluation Details

The human evaluation was conducted by five annotators, including four Ph.D. students and one undergraduate student in computer science. All annotators have strong English proficiency and solid technical backgrounds in machine learning and natural language processing, enabling them to reliably assess question answerability, ambiguity, correctness, and the quality of induced failure-mode descriptions.

Prior to annotation, all annotators were provided with detailed guidelines and calibration examples to ensure consistent interpretation of the evaluation criteria. Each instance was independently annotated by all five annotators.

We show some of the human evaluation interfaces in Figure 14 and Figure 15.

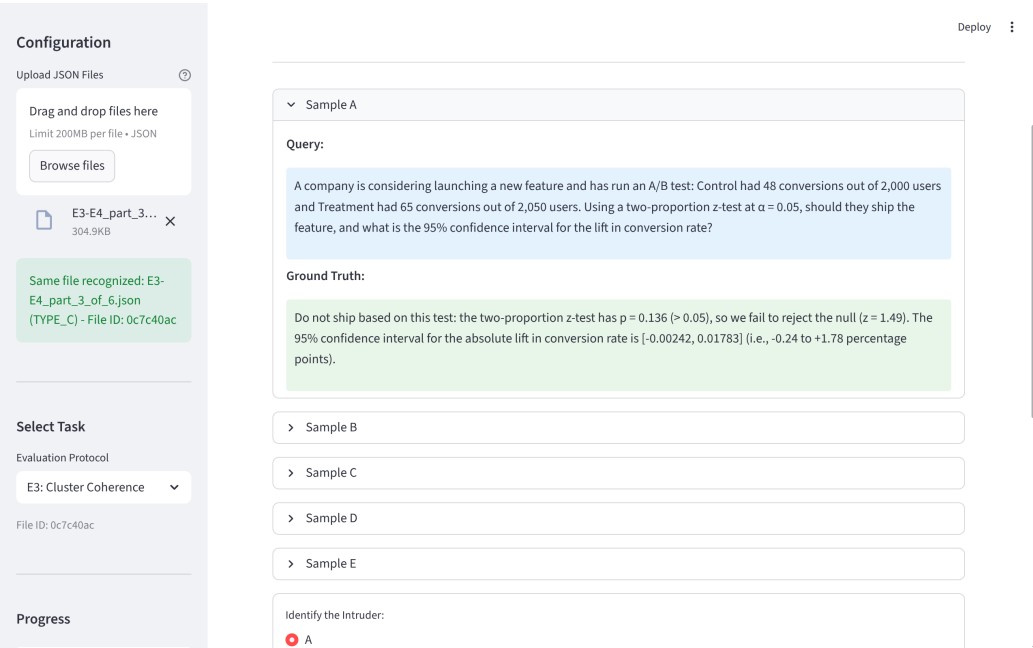

*Figure 15.* Human evaluation interface (2).

## G. Extensibility and Custom Tool Integration

PROBELLM is designed as a modular framework that allows researchers to adapt the probing process to specific domains by integrating custom tools. The implementation decouples the search algorithm from the domain-specific execution logic via a pluggable tool registry based on the Model Context Protocol. The core of our extensibility lies in the `probellm.tools` module which manages tool registration and invocation through three key components. First, the `ToolSpec` defines the interface exposed to the LLM including the tool name and a JSON Schema of its input parameters. Second, the `LocalMCPTool` executes the actual logic by wrapping external libraries and returning a structured dictionary containing execution results or error states. Third, the `ToolRegistry` maps specifications to handlers and routes execution requests during the MCTS expansion phase. This design allows users to inject domain-specific verification or generation logic without modifying the core search algorithm.

To demonstrate the flexibility of PROBELLM for chemistry-oriented weakness discovery (Mirza et al., 2024), we illustrate how to integrate a chemistry-specific tool from the ChemOrch library (Huang et al., 2025e). We utilize the underlying `mol_to_smiles` function to create a tool that standardizes SMILES strings into specific chemical formats. Since the language model interacts via text, the tool is designed to accept a SMILES string, convert it internally to an RDKit molecule object, and output the reformatted string. We first define the tool specification to help the LLM decide when to invoke this tool during test case generation.

```python
from probellm.tools import ToolSpec

mol_to_smiles_spec = ToolSpec(
    name="mol_to_smiles",
    description="Converts an RDKit molecule to a SMILES string with optional
        stereochemistry and Kekule form.",
    input_schema={
        "type": "object",
        "properties": {
            "smiles_input": {
                "type": "string",
                "description": "Input SMILES string to convert the molecule from"
            },
            "isomeric": {
                "type": "boolean",
                "description": "Whether to include stereochemistry information",
                "default": True
            },
            "kekule": {
                "type": "boolean",
                "description": "Whether to output the Kekule form",
                "default": False
            }
        },
        "required": ["smiles_input"]
    }
)
```

*Listing 1.* Defining the Tool Specification

Next, we implement a handler to bridge the interface with the `rdkit` library while ensuring robust error handling. The handler extracts the string parameter, performs the object conversion and function call, and returns a standardized result dictionary. Finally, we register the tool into the system to make it available for the MCTS pipeline.

```python
from probellm.tools import ToolRegistry, LocalMCPTool
from rdkit import Chem

def mol_to_smiles(mol, isomeric=True, kekule=False):
    if kekule:
        Chem.Kekulize(mol)
```

```python
        return Chem.MolToSmiles(mol, kekuleSmiles=True)
    return Chem.MolToSmiles(mol, isomericSmiles=isomeric)

def mol_to_smiles_handler(arguments):
    try:
        smiles_input = arguments.get("smiles_input")
        isomeric = arguments.get("isomeric", True)
        kekule = arguments.get("kekule", False)

        mol = Chem.MolFromSmiles(smiles_input)
        if mol is None:
            return {"success": False, "error": f"Invalid SMILES: {smiles_input}"}

        result = mol_to_smiles(mol, isomeric=isomeric, kekule=kekule)
        return {"success": True, "smiles": result}
    except Exception as e:
        return {"success": False, "error": str(e)}

# Register the tool for use in the pipeline
registry = ToolRegistry()
registry.register(LocalMCPTool(mol_to_smiles_spec, mol_to_smiles_handler))
```

*Listing 2.* Handler Implementation and Registration

By following this pattern, PROBELLM can be easily extended to other specialized domains to make it a versatile framework for broad weakness discovery.

## H. Examples of Discovered Failure Modes

In this section, we illustrate the granularity and interpretability of the structured weaknesses identified by PROBELLM. Unlike automated methods that simply enumerate isolated error cases, our framework induces coherent failure modes that explain *how* and *why* a model fails within a specific domain. We present two failure modes below, formatted to highlight the transition from high-level pattern description to concrete evidence.

### H.1. Mode: Schema Design Constraint Violation

---

**Cluster 7: Constraint Drift in Relational Schema Design**

This failure mode emerges in database modeling tasks requiring adherence to normalization principles and integrity constraints. The model generates plausible schema structures but systematically violates fundamental design rules that prevent data anomalies. Specifically, it reintroduces redundancy by storing denormalized attributes—such as placing `author` directly in the `Book` table while simultaneously defining separate `Author` and `BookAuthor` junction tables—thereby contradicting the stated decomposition goal. Additionally, proposed CHECK constraints frequently reference subqueries or other tables, which violates SQL standard limitations and renders the constraints unenforceable in most relational database systems. The root cause appears to be pattern-matching against common schema templates without verifying constraint semantics or rigorously applying normalization rules to eliminate redundancy.

- - - - - - - - - - - - - - - - - - - - - - - - - - - - - - - - - - - - - - - - - - - - - - - - - - - - - - - - - - - - -

**Example Instance**

*Query:* "Design a normalized relational database schema for a public library system... justify how your design prevents update anomalies."

*Ground Truth Schema (excerpt):* `BOOK(book_id PK, isbn, title, publisher); AUTHOR(author_id PK, name); BOOK_AUTHOR(book_id FK, author_id FK, PK(book_id, author_id));`

*Model Prediction (excerpt):* `Book(book_id PK, isbn, title, `**`author`**`, publisher...); Author(author_id PK, full_name...); BookAuthor(book_id FK, author_id FK, PK...);`

*Diagnostic Analysis:* The model schema retains an `author` column inside the `Book` table despite also defining a normalized `Author` junction table. This reintroduces redundancy and update anomalies—changing an author's name would require updating both tables inconsistently. Furthermore, two of the model's proposed CHECK constraints embed subqueries with `NOT EXISTS` clauses referencing other tables, which standard SQL disallows in CHECK constraints, making them syntactically invalid.

---

## H.2. Mode: Bitwise Arithmetic Integrity

---

**Cluster 3: Bitwise Pipeline Slip & Inconsistent Verification**

We identify a distinct failure mode in the domain of low-level arithmetic and cryptographic primitives, specifically probing multi-step 32-bit operations. The failure manifests as "pipeline slips," where a minor calculation error in one stage—such as an off-by-one rotation or incorrect carry handling—propagates through subsequent XOR masks and modular additions, resulting in a completely incorrect final value. A critical feature of this mode is the inconsistency between generation and verification: the model often generates a self-verification step that contradicts its own previous derivation or implicitly assumes undefined function behaviors. This suggests a systematic weakness in maintaining deterministic bookkeeping for 32-bit boundary conditions and a tendency to hallucinate function definitions rather than halting execution.

- - - - - - - - - - - - - - - - - - - - - - - - - - - - - - - - - - - - - - - - - - - - - - - - -

**Example Instance**

*Query:* [Complex bit manipulation involving `ROTR`, `XOR`, and modular addition steps...]

*Ground Truth:* [Specific 32-bit Hex Value ending in ...A3]

*Model Prediction:* [Incorrect Hex Value ending in ...B4]

*Diagnostic Analysis:* The model fails to maintain state consistency across multiple reversible operations. Although the individual arithmetic steps appear plausible, the accumulated error leads to a verification failure where the inverse function does not reconstruct the original input, yet the model outputs the result confidently.

---

