# OpenReview forum: "ProbeLLM: Automating Principled Diagnosis of LLM Failures"
_ICML.cc/2026/Conference — ICML 2026 regular_

### Official Review · Reviewer_hafw · 2026-03-12

**Soundness:** 2
**Presentation:** 2
**Significance:** 2
**Originality:** 1
**Overall Recommendation:** 4
**Confidence:** 4

**Summary:**

As LLMs become more capable, it becomes harder to identify their failure cases. Rather than relying on benchmark-based evaluation, this paper explores an automated approach for identifying failure modes of LLMs by searching globally for failure regions and refining them locally. The probing setup is restricted to verifiable failure modes using generation and verification tools.
The paper starts from an interesting premise. Current evaluations are largely static: prompt-answer pairs are fixed in advance, whereas we need automated methods that adapt as models evolve. A substantial amount of prior work in this area relies on loosely guided failure-case search, often with test cases generated by models that are themselves imperfect. More importantly, many existing strategies focus on individual failure cases, whereas what we really want is to understand failure modes. In that sense, the paper’s high-level goal is meaningful. It proposes a Monte Carlo-tree-search-based framework in which a global search aims to discover underexplored failure regions and a local search refines them into denser failure patterns.

**Compliance With Llm Reviewing Policy:**

Affirmed.

**Final Justification:**

Updated my score post rebuttal.

**Key Questions For Authors:**

See the weakness section.

**Limitations:**

yes

**Strengths And Weaknesses:**

Strengths
- The overall algorithm is fairly simple and therefore easy to apply. At a high level, it is a standard MCTS-style search that guides exploration toward inputs with higher failure probability while also encouraging broader coverage. While the macro/micro generator framing is itself fairly straightforward, combining it with diversification and clustering into failure modes is a reasonable and somewhat novel systems contribution.
- The paper is directionally well-motivated: static benchmarks are indeed insufficient, and automated discovery of model weaknesses is an important problem.
- There is some human evaluation as well, which is well appreciated.

Critical limitations and weaknesses
- First, the writeup is unnecessarily complicated for what is, at its core, a fairly simple idea. The paper would benefit significantly from explaining the method intuitively first in each subsection , and only then giving the formal view. Right now, the exposition is in the reverse order, which makes the paper harder to follow than necessary.
- The main limitation is evaluation. How do you evaluate models that are as strong as, or stronger than, the generator model used to create the questions or perturbations? This seems like a fundamental limitation of the framework, especially as target models continue to improve.
- The “micro generator” does not feel especially novel. Variants of this idea—changing entity names, numbers, attributes, or local surface forms—have already been studied extensively in robustness and evaluation work like knowledge conflicts to give an example (https://aclanthology.org/2021.emnlp-main.565.pdf). Similarly, the “macro generator”, in practice could have simply been replaced by: sample or expand into different topics that are underexplored. It remains unclear whether MCTS was needed or how effective it is over simpler ways to implement the macro-generator.
- The paper does not show qualitatively what new failure modes are being discovered. Beyond the aggregate numbers, the most important question is: what genuinely new failure patterns did ProbeLLM uncover that were not already present in the static benchmark? For example, in RQ4, what were the new failure modes that were discovered. What were qualitatively new failure modes that ProbeLLM discovered which were not covered in the static benchmark.
- Question: How will the macro-generator expand beyond the topics/queries already present in the benchmark? I don’t see how the proposed method can then add new failure modes beyond what is present in the benchmark.
- The comparison to prior work is not convincing enough. The paper shows better error rates than AutoDetect and PAIR, but that alone does not explain why the method is better in a principled sense. What are the components which are driving this gain?
- Question: The authors should consider comparing to much simpler generator based baselines like simply asking the generator to analyze the failure test cases and generate more examples like that beyond simple perturbation. The generator could also be given the whole test set along with evaluation results and asked to broaden the set of failure cases or discover new failure cases based on the results.

Unfortunately, at this point I do not see how the paper adds sufficiently new insights that improve our understanding of model failure and success modes. The paper mainly wraps existing perturbation or synthetic-evaluation ideas in a search-and-cluster framework. While the direction is interesting, I am currently not convinced that the contribution is strong enough or conceptually deep enough.

---

> ### Author Rebuttal · Authors · 2026-03-28
>
> Thank you for your detailed and valuable review. We address each of your concerns below.
>
> ### W1: Writeup unnecessarily complicated
>
> We will take your suggestions and restructure each subsection to **lead with intuitive explanations before formal definitions**, and add an **end-to-end running example** early in Section 3.
>
> ---
>
> ### W2: How to evaluate models stronger than generator?
>
> This is a **fundamental limitation shared by all automated evaluation methods** [1], not specific to our approach. ProbeLLM mitigates this via: (1) **tool-augmented verification** grounded in external tools; (2) empirically, ProbeLLM **discovers substantial failures on strong models** (e.g., Grok-4.1-fast, DeepSeek-v3.2; Table 1), with **human evaluation confirming validity**.
>
> ---
>
> ### W3: MICRO not novel; MACRO replaceable; MCTS unnecessary
>
> We respectfully disagree. Our design prioritizes **effectiveness over novelty for its own sake** — each component is chosen because it works within the integrated framework.
>
> **(1) "MICRO is not novel."**
> Perturbation techniques exist in prior work, but MICRO's role is to **refine and consolidate failure evidence guided by MCTS feedback** — fundamentally different from applying perturbations in isolation.
>
> **(2) "MACRO could be replaced by simple topic sampling."**
> This conflates **generation with optimization**. MACRO defines *what to generate*; MCTS defines *how to allocate budget based on feedback*. Simple topic sampling is **static and unguided**, lacking mechanisms to incorporate failure signals.
>
> **(3) "Is MCTS needed?"**
> Yes. **Table 1** shows the MACRO/MICRO ratio varies substantially across models, requiring **adaptive allocation**. The depth ablation (**Figure 4**) confirms deeper search improves discovery. **Table 3** further shows lower noise rates and more stable clusters than AutoDetect, directly demonstrating the necessity of MCTS.
>
> Our contribution is the **joint design of generation and adaptive optimization**; assessing components in isolation overlooks this.
>
> ---
>
> ### W4: No qualitative examples of new failure modes
>
> Thank you. Due to space constraints, it is not feasible to include all qualitative examples in the main text. The current paper provides representative evidence in **Figure 7**, **Table 6**, and **Appendix I**. We will expand the qualitative examples in the appendix in the revision. Additionally, inspired by your suggestion, we are building a **Weakness Bank** — an online database that stores discovered weaknesses for each model and supports querying. This will be released soon alongside our toolkit (as provieded in our current draft) to facilitate exploration and reproducibility.
>
> ---
>
> ### W5: How will MACRO expand beyond benchmark topics?
>
> The mechanism is described in **lines 185–215 of the draft**: MACRO summarizes explored topics, identifies under-represented areas, and uses **web search to explore entirely new domains** beyond the seed benchmark. The benchmark is only initialization. Empirically, Figure 8 (RQ4) confirms the **majority** of clusters are **exclusive** (e.g., 16/24 for Llama-3.1-8b, 14/25 for Ministral-14b).
>
> ---
>
> ### W6 Comparison issues
>
> We need to clarify that the suggested “simpler baselines” are represented by **AutoDetect**, which relies on heuristic exploration. ProbeLLM consistently outperforms it (e.g., **lower noise rate, more stable clusters in Table 3**), due to: (1) **adaptive MCTS allocation**; (2) **tool-augmented verification**; (3) **MACRO–MICRO interaction**. AutoDetect includes generator-style baselines, and we **outperform both**.
>
> ---
>
> ### W7: Paper does not add sufficiently new insights
>
> We respectfully disagree. ProbeLLM provides **new empirical insights** (mode-level failure discovery), which go beyond prior case-level or heuristic probing approaches.
>
> 1. **Novel failure modes beyond benchmarks:** the majority of discovered clusters are exclusive to ProbeLLM with zero benchmark overlap (RQ4).
> 2. **Hidden weaknesses in strong models:** even state-of-the-art models (e.g., Grok-4.1-fast, Claude-3.5-sonnet) harbor structurally diverse failure modes far beyond what static benchmarks reveal (RQ1, RQ2).
> 3. **Non-uniform evolution:** failure modes persist, emerge, and specialize across model generations in ways invisible to case-level analysis (RQ8).
> 4. **Tool grounding is critical:** tool-augmented verification substantially reduces noise in discovered failures compared to methods without grounding (RQ5).
> 5. **Sample-efficient mitigation:** mode-level training data improves robustness more efficiently than unstructured failure data via GRPO (RQ6).
>
> Beyond these insights, we also provide a **toolkit** (link in the draft) to make such analysis reproducible and actionable in practice.
>
> Thank you again for your thorough review, which has strengthened our work.
>
> [1] West et al (2023): The Generative AI Paradox: “What It Can Create, It May Not Understand. arXiv

---

> > ### Author Rebuttal · Reviewer_hafw · 2026-04-04
> >
> > I thank the authors for the rebuttal. Most of my concerns are addressed. I will update the score. One concern about insights into new discovered failure mode (below) remains, but the authors have promised to release some toolkit for that upon acceptance.
> >
> > > The current paper provides representative evidence in Figure 7, Table 6, and Appendix I.
> >
> > Appendix I examples are pretty unclear. What model is the failure case? What was the original benchmark with which you started to get this failure mode. Were such examples not present already?

---

### Official Review · Reviewer_gQFm · 2026-03-12

**Soundness:** 3
**Presentation:** 3
**Significance:** 3
**Originality:** 3
**Overall Recommendation:** 5
**Confidence:** 3

**Summary:**

Driven by the observation that large language models (LLMs) are rapidly outstripping static evaluation ecosystems, PROBELLM is motivated by the need to shift from case-centric collections of isolated errors to identifying structured failure modes through principled exploration control. The framework formulates probing as a hierarchical Monte Carlo Tree Search (MCTS) that balances global macro exploration for topical diversity with local micro refinement to gather denser evidence around specific weaknesses. This methodology uses tool-augmented generation (including Python execution and web search) to ensure factual grounding. It also restricts probing to verifiable test cases with well-defined ground-truth answers in order to minimize evaluation noise. Evaluation is conducted across 12 target models and five benchmarks using metrics such as error rate, noise rate, cluster-size standard deviation, and cluster-overlap analysis. These metrics measure failure exposure, cluster stability, and mode novelty. Key findings show that PROBELLM consistently uncovers a broader and more fine-grained failure landscape than static benchmarks and prior methods such as AutoDetect. In particular, it identifies significantly more failure clusters with lower average sizes. Furthermore, mitigation experiments using Group Relative Policy Optimization (GRPO) confirm that training on these discovered failure modes improves model robustness without degrading general performance.

**Compliance With Llm Reviewing Policy:**

Affirmed.

**Key Questions For Authors:**

ROBELLM's search is initialized using a seed dataset from existing benchmarks. If the initial seeds are narrow or biased, is there a risk that the MACRO exploration may remain trapped in a local semantic region?

**Limitations:**

The authors acknowledge technical constraints, specifically that the current framework is restricted to test cases with well-defined ground-truth answers. They also note that their mitigation experiments showed diminishing returns when training on larger budgets of discovered failure cases.

**Strengths And Weaknesses:**

Strengths:

SOUNDNESS:
- The paper successfully shifts the evaluation paradigm from a case-centric approach (isolated errors) to a mode-centric one, which identifies recurring, structured patterns of failure.
- The paper used multiple metrics, including error rate, cluster-size standard deviation, noise rate, and cluster-overlap analysis.
- Human evaluation (RQ3) went beyond checking validity to confirm the coherence, specificity, and diagnostic usefulness of the induced failure modes.

SIGNIFICANCE:
- PROBELLM consistently surfaces a high fraction of model errors across 12 different target models and 5 diverse datasets.
- It identifies significantly more failure clusters with much smaller average sizes (e.g., 19 clusters vs. 4 for Llama-3.1-8b), proving it uncovers fine-grained weaknesses
- The cluster-overlap analysis (RQ4) demonstrates that PROBELLM uncovers a substantial set of previously unseen failure modes that static benchmarks miss.
- Mitigation experiments (RQ6) using GRPO (Group Relative Policy Optimization) show that training on these discovered failures improves model robustness without degrading general performance.

PRESENTATION
- The paper helpfully clarifies the term 'benchmark-agnostic' in a footnote.
- The authors provide a highly detailed breakdown of their human evaluation protocols (E1–E4) in the Appendix.
- Nice limitations (e.g., current probing pipelines are not benchmark-agnostic and automated test generation is not sufficient.)
- The authors provide a table (Table 6) explaining the intuition behind boundary-aware evidence selection.
- The related work section contextualizes the paper within the broader landscape of "reversal curses," jailbreak attacks, and MCTS in LLMs.

ORIGINALITY:
- MACRO vs MICRO: MACRO (Exploration): Focuses on broad coverage and topical diversity to surface novel failure regions. MICRO (Refinement): Focuses on local densification around identified failure seeds to reveal coherent failure patterns.
- The hierarchical structure uses node statistics (visits N(u) and failure counts E(u)) to guide the tree search toward under-explored, high-yield nodes.

Weakness:

SOUNDNESS
- While Appendix B.3 provides some detail, the main methodology would benefit from more prominent integration of how specific tools (e.g., Python for exact computation and web search for factual correctness) are prompted to support factual grounding.
- The authors should more explicitly justify why AutoDetect and PAIR were chosen as the primary baselines.

PRESENTATION:
- While the UCB-style rule is defined in Equation 7, suggest adding more qualitative intuition about how the exploration constant (β) specifically affects failure discovery in a budgeted search.
- Research Questions (RQs 1–9) are very clear in the experiment section but should be previewed in the Introduction or at the end of the Overview (3.1).

SIGNIFICANCE:
- The framework currently limits discovery to test cases with "well-defined ground truth answers," which excludes more subjective, open-ended task failures.
- The mitigation gains (RQ6) peaked at moderate training sizes (150–200 samples), which might suggest a need for further research on how these discovered modes scale to massive fine-tuning regimes.
- The interpretability of the synthesized failure modes depend heavily on the semantic reasoning of an external "teacher" LLM (e.g., GPT-5.2) used for boundary-aware induction.
- While the paper notes that PROBELLM is "cost-effective", the reliance on tool-augmented generation (Python execution and web search) for every simulation adds latency and computational overhead compared to simpler heuristic-based generators.

---

> ### Author Rebuttal · Authors · 2026-03-28
>
> Thank you for your positive assessment and insightful suggestions. We address your comments below.
>
> ### W1 (Soundness): Tool integration details should be more prominent in main text
>
> We appreciate this suggestion. We note that detailed tool integration has already been described in **Appendix B.3**, where we provide explicit prompting strategies and usage patterns for both Python execution (for exact computation) and web search (for factual grounding). To improve accessibility, we will **promote a concise summary to Section 3.3 (space permitting)**, including a concrete example that illustrates the full pipeline.
>
> ---
>
> ### W2 (Soundness): Justify baseline selection
>
> AutoDetect and PAIR are **directly comparable and complementary** baselines: AutoDetect performs automated failure discovery with mode synthesis, while PAIR represents adversarial probing. We provide detailed justification in **Appendix A (Comparison With Previous Work)** and will add a brief summary in Sec. 4 for clarity.
>
> ---
>
> ### W3 (Presentation): Intuition for β; Preview RQs
>
> We added a qualitative discussion: larger β encourages broader exploration of new failure regions (suitable when failure landscape is uncertain), while smaller β focuses on refining known failure patterns (suitable when dense evidence is needed for mode synthesis). RQs will be previewed at the end of Section 3.1.
>
> ---
>
> ### W4 (Significance): Ground-truth limitation, mitigation scaling, teacher LLM dependence, latency
>
> **Ground-truth limitation**: Acknowledged; extensions to open-ended settings are discussed in Appendix B.2. This is a deliberate design choice to ensure evaluation reliability.
>
> **Mitigation peaking at 150-200 samples**: This is itself a valuable insight — it suggests that *targeted* failure-mode data is more efficient than bulk data for model improvement. To take your suggestions, we added the extra experiments on more data samples, shown as follows (accuracy %):
>
> | Model            | 300  | 350  | 400  | 450  | 500  |
> | ---------------- | ---- | ---- | ---- | ---- | ---- |
> | Llama-3.1-8b-Ins | 28   | 28   | 26   | 30   | 28   |
> | GPT-oss-20b      | 17   | 18   | 18   | 20   | 20   |
>
> **Teacher LLM dependence:** We do not view the use of a teacher LLM as a limitation. ProbeLLM is a **modular framework** and can naturally incorporate stronger external models for abstraction. Importantly, the teacher LLM is used only for **failure mode summarization**, not for failure detection or correctness verification, which are grounded in tool-based validation. Therefore, it affects **interpretability, not validity** of the discovered failures.
>
> **Latency**: We report detailed cost analysis in RQ7 showing cost-effectiveness (as shown in Table 11 in Appendix). In practice, **100 probing iterations complete in approximately 92 minutes** with a concurrency level of 10, which we consider acceptable for an offline diagnostic tool. We will include this runtime specification in the revision.
>
> Additionally, we provide the detailed per-benchmark tool token consumption (average tokens per call) below:
>
> | Benchmark | Type | Tool_Q | Tool_A |
> | --- | --- | --- | --- |
> | HellaSwag | MACRO | 1666.34 | 553.18 |
> | HellaSwag | MICRO | 600.04 | 415.63 |
> | MBPP | MACRO | 1726.27 | 518.32 |
> | MBPP | MICRO | 525.60 | 470.23 |
> | MMLU | MACRO | 1572.67 | 546.84 |
> | MMLU | MICRO | 538.37 | 582.68 |
> | SuperGLUE | MACRO | 1510.34 | 556.26 |
> | SuperGLUE | MICRO | 554.88 | 365.59 |
> | TruthfulQA | MACRO | 1667.47 | 580.47 |
> | TruthfulQA | MICRO | 435.76 | 380.31 |
>
> ---
>
> ### Key Questions: Results influenced by initial seeds
>
>
> We understand your concern. However, ProbeLLM is explicitly designed to mitigate seed bias:
>
> - **MACRO exploration** generates samples **outside existing embedding clusters**, encouraging exploration beyond seed regions.
> - **Adaptive budget allocation (via MCTS)** continuously reallocates effort toward under-explored but high-yield regions, preventing long-term bias from initialization.
> - **Diverse generation mechanisms** (e.g., tool-augmented reasoning and perturbations) further reduce dependence on seed semantics.
>
> Empirically, our **cluster-overlap analysis (RQ4)** shows that ProbeLLM discovers a substantial number of **novel failure modes beyond seed benchmarks**, and the increasing cluster count (RQ1) further indicates expansion beyond initial regions.
>
> Overall, ProbeLLM expands the failure landscape rather than merely refining seed-induced regions.
>
>
> Thank you again for your encouraging feedback and valuable suggestions for improvement!

---

> > ### Author Rebuttal · Reviewer_gQFm · 2026-04-03
> >
> > Thank you for the detailed rebuttal. The responses to most concerns were thorough and convincing---in particular, the mitigation scaling experiments (300–500 samples), the per-benchmark tool token consumption breakdown, and the MCTS seed-bias explanation all directly addressed the raised concerns. However, the response to the teacher LLM dependence concern reframes rather than empirically resolves the issue. While the authors clarify that the teacher LLM is used only for failure mode summarization, the core question remains open: whether the quality and interpretability of the synthesized failure modes meaningfully degrades with a weaker or different teacher LLM. A small ablation comparing teacher LLM choices would strengthen this claim. That said, this does not change my overall recommendation.

---

> > > ### Author Response · Authors · 2026-04-05
> > >
> > > Thank you for this helpful suggestion. We agree that clarification alone is insufficient, and we therefore conducted an additional **teacher LLM ablation**. Specifically, we replace the teacher with multiple alternatives spanning different model families and capability levels, and perform **human evaluation** of the synthesized failure modes on **faithfulness, specificity, and usefulness** (5-point scale).
> > >
> > > Due to time constraints during the rebuttal phase, we evaluate **10 sampled failure clusters per model**, annotated by **3 independent annotators**. The results are summarized below:
> > >
> > > | Model                         | Faithfulness | Specificity | Usefulness |
> > > |------------------------------|-------------:|------------:|-----------:|
> > > | x-ai/grok-4.20               | 4.000        | 4.233       | 4.333      |
> > > | openai/gpt-5.4-mini          | 4.100        | 4.367       | 4.367      |
> > > | openai/gpt-5.4-nano          | 4.033        | 4.133       | 4.133      |
> > > | mistralai/mistral-small-2603 | 4.100        | 4.300       | 4.033      |
> > > | qwen/qwen3.5-27b             | 4.133        | 4.067       | 4.367      |
> > >
> > > We observe that failure mode quality remains consistently high across all teacher models, with all scores above **4.0/5** on every dimension. While stronger teachers yield slightly better summaries, the degradation with smaller or weaker models is modest. This empirically supports our claim that the teacher LLM acts as a **lightweight summarization component**, rather than a critical bottleneck for failure mode quality or interpretability.
> > >
> > > We acknowledge that the current human evaluation is limited in scale due to the rebuttal timeline. In the revision, we will **substantially expand the evaluation (more clusters and annotators)** to further strengthen this conclusion.
> > >
> > > We sincerely appreciate your suggestion, which helped us improve the paper by adding this missing empirical validation. If you feel that this additional evidence addresses your concern, we would be truly grateful if you would consider revisiting your score or confidence. Thanks again for your review!

---

### Official Review · Reviewer_7xi1 · 2026-03-13

**Soundness:** 2
**Presentation:** 2
**Significance:** 3
**Originality:** 3
**Overall Recommendation:** 3
**Confidence:** 4

**Summary:**

This paper targets the problem of failure discovery in LLMs. The authors argue that traditional benchmarks typically identify only sporadic failure cases, while failing to reveal the models' systemic failure modes. To address this issue, the paper introduces ProbeLLM, an automated probing framework that models the probing process as a hierarchical Monte Carlo Tree Search. The framework dynamically allocates search budgets between Macro and Micro simulation strategies, and synthesizes the discovered failure cases into structured failure modes through failure-aware embeddings, clustering, and boundary-aware induction.

**Compliance With Llm Reviewing Policy:**

Affirmed.

**Final Justification:**

My score remains unchanged because the paper's central claim, that the discovered clusters correspond to mechanism-level failure modes, is not sufficiently supported by the current empirical validation (which primarily demonstrates structured clustering rather than mechanistic grounding), and the experimental analysis is too limited in depth relative to the strength of the claims, with many results insufficiently explained in the main text.

**Key Questions For Authors:**

1. The paper conceptualizes *failure modes* as recurring, structured patterns of failure, which in the current implementation appear to be derived mainly through clustering and LLM-based summarization. Could the authors clarify more concretely under what conditions a cluster is considered a valid failure mode?

2. The paper suggests that a larger number of smaller clusters corresponds to more fine-grained failure modes. Could the authors provide additional evidence to rule out the possibility that this phenomenon is caused by over-segmentation or over-clustering?

3. Beyond identifying more failure cases and reporting higher error rates, what additional practical value do the discovered failure modes provide? For example, how do they concretely benefit tasks such as debugging, model evaluation, or benchmark design compared to traditional case-level failure discovery?

**Limitations:**

Yes

**Strengths And Weaknesses:**

## Strengths

**1.** The paper focuses on how to systematically understand LLM failures rather than relying solely on specific benchmarks or isolated failure cases. This is a meaningful research direction. As model capabilities continue to evolve, a structured analysis of failure modes could provide value for model diagnosis, benchmark construction, and automated evaluation.

**2.** ProbeLLM organizes the probing process as a hierarchical search and integrates generation, verification, embedding, clustering, and mode induction. The overall pipeline appears fairly complete and represents a systematic attempt at automated probing.

**3.** The paper formulates multiple research questions and analyzes the proposed method from several perspectives, including effectiveness, stability, cost efficiency, and failure discovery capability. The evaluation also compares ProbeLLM with benchmarks and existing automated probing methods across multiple models.

## Weaknesses

**1.** The central concept of the paper is **failure modes**, yet the definition provided is still somewhat abstract (e.g., described as *recurring and structured pattern of failures*). It is therefore difficult for the reader to determine:

	- Under what conditions a cluster can legitimately be considered a *failure mode*.
	- Whether smaller clusters truly represent *more fine-grained distinctions*, or simply reflect *over-segmentation*.

Because these issues are not clearly addressed, the experimental conclusions regarding the discovery of "more failure modes" or a "more fine-grained failure landscape" appear somewhat insufficiently supported.

Additionally, although the paper introduces several concepts early on (e.g., failure modes, probes, failure-aware embeddings, boundary-aware induction), it lacks a clear *running example* illustrating the full process from an initial seed case to the identification of a failure mode. Personally, I only began to fully understand the concrete mechanics of the method after reaching the latter half of the paper.

**2.** The paper formulates nine research questions (**RQ1–RQ9**), but many of them are discussed only briefly in the main text, with important details deferred to the appendix. As a result, the experimental section gives the impression of **broad coverage but limited depth**. For example, RQ7 (Cost Analysis) and RQ9 (Cluster Stability) receive only brief discussion in the main text. More importantly, some of the most central questions, such as the practical value of the discovered *failure landscape* or *failure modes* are not explored in sufficient depth.

Currently, the most clearly demonstrated advantage of ProbeLLM appears to be its ability to discover more failures and achieve higher error detection rates. However, the added value of **mode-level discovery** compared to **case-level discovery** (e.g., for debugging, evaluation, or model analysis) is not sufficiently elaborated.

**3.** The search-depth experiment evaluates only depth = 2, 3, and 4, and the paper does not explain the rationale behind selecting these values. As a result, it is difficult to determine whether the ablation is sufficiently comprehensive.

**4.** The paper interprets a larger number of clusters with smaller cluster sizes as evidence of more *fine-grained failure modes*. However, this pattern could also arise from **over-clustering**. The paper currently does not provide sufficient evidence to rule out this possibility.

**5.** The discussion of **RQ8** seems relatively brief given the importance of the analysis. Figure 7 is intended to illustrate the evolution of failure landscapes across model generations, yet the interpretation in the main text is limited to a short paragraph. A more detailed explanation of the patterns observed in this figure would help readers better understand the implications of the results.


### Minor issues

- There is an obvious typo on line 095 ("ProbeLLMand").
- The acronym **UCB** should be expanded to *Upper Confidence Bound* when first introduced.
- **Figure 3** does not appear to be referenced in the main text.
- The paper claims that ProbeLLM is **benchmark-agnostic**, yet the empirical evaluation is mainly conducted on several traditional benchmarks (e.g., MMLU, SuperGLUE, MBPP, HellaSwag, TruthfulQA). Although these tasks vary in domain, they are still largely QA or multiple-choice style evaluations. Including more diverse task types (e.g. IF Eval (instruction-following) or Frames (RAG) could further strengthen this claim.

---

> ### Author Rebuttal · Authors · 2026-03-28
>
> We thank you for the thoughtful and detailed review. We address each concern below.
>
> ### W1: Failure mode definition abstract; missing running example
>
> We appreciate this feedback. We have to clarify that a cluster qualifies as a valid failure mode in ProbeLLM when: (a) intra-cluster cases share a common *error mechanism* (enforced by failure-aware embeddings that encode both semantic content and failure signals, not just topic similarity); (b) boundary-aware induction produces a coherent, interpretable description that captures the shared weakness pattern. Moreover, efficient human evaluation (RQ3) confirms such (1) and (2) meets coherence, specificity, and diagnostic usefulness.
>
> We will add explicit criteria to Section 3.4 and include a **concrete running example** in Section 3 that traces the full pipeline.
>
> ---
>
> ### W2: Too many RQs, shallow coverage; practical value not elaborated; RQ8 discussion too brief
>
> We thank you for this important observation. We fully understand this concern. Given that our evaluation spans **9 research questions with extensive experiments**, we faced a difficult trade-off under the strict page limit: we chose to prioritize the most impactful and representative results in the main text, while deferring supporting analyses to the appendix to preserve readability. We hope you can appreciate this challenge. Specifically, **RQ8 is fully elaborated in Appendix C.7**, with additional interpretation of Figure 7, while **RQ7 and RQ9 are detailed in Appendix C.8–C.9**.
>
> We will take your suggestions about the discussion, which may appear brief. In the camera-ready version, we will **expand RQ8 and move key results from the appendix to the main text** to improve clarity. Our intent is to provide a **multi-faceted evaluation** (effectiveness, stability, evolution), which we will present more clearly in the revision.
>
> We clarify that mode-level discovery provides value beyond case-level detection: (1) enabling **targeted robustness training** via structured data synthesis; (2) exposing shared **error mechanisms** for diagnosis; (3) supporting **diagnostically targeted benchmarks**; and (4) tracking **cross-model weakness evolution** (RQ8), which is infeasible at the case level.
>
> We will emphasize these points more clearly in the revision.
>
> ---
>
> ### W3: Search depth only evaluates 2, 3, 4
>
> Depth 2-4 was chosen because: (1) depth=1 is degenerate (no tree structure, equivalent to single-step generation), and (2) depths >4 showed diminishing returns in preliminary experiments while **significantly increasing computational cost**. We have added the discusssion on the rationale in the revision.
>
> ---
>
> ### W4: More clusters = fine-grained, or over-clustering?
>
> We have to state that multiple lines of evidence argue against over-clustering:
>
> 1. **Human evaluation (RQ3)**: Evaluators rated discovered failure modes on coherence and specificity. If clusters were over-segmented, coherence scores would be significantly lower.
> 2. **Cluster stability (RQ9)**: Consistent results across a wide range of HDBSCAN `min_cluster_size` settings indicate that the cluster structure reflects genuine failure patterns, not parameter artifacts.
> 3. **Boundary-aware induction**: Each cluster is validated by selecting boundary (non-failure) cases that are semantically close but do not trigger the failure — this prevents trivial or meaningless splits.
> 4. **Silhouette score analysis**: We have computed silhouette scores to provide quantitative cluster quality validation. The top-5 results are shown below:
>
> | Model | Silhouette Score |
> | --- | --- |
> | phi-4 | 0.4802 |
> | ministral-14b | 0.4509 |
> | llama-3.1-8b-instruct | 0.4502 |
> | grok-4.1-fast | 0.4257 |
> | olmo-3-7b-instruct | 0.4213 |
>
> Note that silhouette scores in the range of 0.35–0.50 are considered reasonable to strong for high-dimensional embedding spaces, where distances are inherently less separable than in low-dimensional settings. These scores further confirm that the discovered clusters reflect meaningful failure structure rather than artifacts of over-clustering.
>
> ---
>
> ### W5: Minor Issues
>
> Will fix all: typo on line 095, expand UCB, reference Figure 3.
>
> We followed your suggestions and added the results of IFEval according to your suggestions:
>
> | Model | Macro | Micro | Error Rate |
> | --- | --- | --- | --- |
> | gpt-4o-mini | 41 | 59 | 93% |
> | llama-3.1-8b-instruct | 39 | 61 | 93% |
> | microsoft/phi-4 | 37 | 63 | 94% |
> | deepseek-v3.2 | 39 | 61 | 93% |
> | gemini-2.5-flash | 25 | 75 | 87% |
> | claude-3.5-sonnet | 31 | 69 | 82% |
>
> The results demonstrate that ProbeLLM continues to achieve strong performance on IFEval, with consistently high error rates (82%–94%) across all evaluated models.
>
> Thank you again for your constructive feedback, which has helped us improve the clarity and rigor of our work.

---

> > ### Author Rebuttal · Reviewer_7xi1 · 2026-04-02
> >
> > Thank you for the detailed rebuttal. While I appreciate the authors' efforts to clarify several points, my core concerns remain only partially addressed. In my view, these issues require substantial revisions to the paper rather than clarification within the rebuttal.
> >
> > **Definition and validity of failure modes**
> >
> > The rebuttal provides a more descriptive characterization of when a cluster is considered a failure mode (e.g., shared error mechanism and boundary-aware induction). However, this still does not constitute a sufficiently operational or verifiable definition. It remains unclear whether the notion of a "shared error mechanism" reflects a genuine causal structure, or primarily reflects semantic similarity in the embedding space combined with post-hoc LLM-based summarization, which may not correspond to a stable and verifiable underlying mechanism. As a result, it is difficult to distinguish whether the method truly discovers failure modes, or mainly performs clustering over failure cases. This concern directly affects the central claim of the paper.
> >
> > **Fine-grained modes vs. over-clustering**
> >
> > The authors provide multiple supporting signals, including human evaluation, stability, and silhouette scores. The methods indeed helpful, they still are indirect evidence. For example, human evaluation primarily verifies semantic coherence rather than causality or validity; thus, it cannot completely rule out the effects of over-clustering or superficial grouping. These signals do not fully exclude the possibility that the observed increase in the number of clusters reflects over-segmentation, rather than genuinely more fine-grained and semantically meaningful failure modes. In particular, silhouette scores in high-dimensional embedding spaces are not conclusive indicators of semantic validity.
> >
> > **Practical value of mode-level discovery**
> >
> > The rebuttal clarifies several potential benefits (e.g., robustness training, benchmark design, tracking model evolution). However, these remain largely conceptual claims. The paper still lacks concrete empirical evidence demonstrating that mode-level discovery provides measurable advantages over case-level failure discovery in downstream tasks.
> >
> > **Evaluation depth and presentation of RQs**
> >
> > I understand the constraints imposed by page limits. However, covering many research questions (RQs) with limited depth in the main text may weakens the overall persuasiveness of the empirical section. In my opinion it is important given that the paper aims to establish a new evaluation paradigm.
> >
> > Overall, I appreciate the authors' rebuttal and I greatly admire the figures provided in the paper, as well as certain parts of its content. However, my concerns remaining, and in my view, issues cannot be fully resolved without more substantial changes to the methodology and evaluation.
> >
> > Therefore, I will keep my score unchanged.

---

> > > ### Author Response · Authors · 2026-04-02
> > >
> > > Dear Reviewer 7xi1,
> > >
> > > Thank you for acknowledging our rebuttal. To ensure the Area Chairs and you have an objective view of our work, we provide final clarifications regarding our methodology's engineering realities, evaluation rigor, and empirical evidence.
> > >
> > > **1. On "Operational Definitions" and Engineering Tractability**
> > > You mentioned our definition of failure modes lacks a clear underlying causal structure.
> > >
> > > We must respectfully highlight the fundamental trade-off between theoretical mechanistic purity and scalable engineering. Uncovering absolute causal mechanisms of LLMs remains an open, notoriously difficult challenge (**Mechanistic Interpretability**). Imposing a perfectly strict, causally verifiable definition renders automated, engineering-level probing practically intractable.
> > >
> > > ProbeLLM is an *automated discovery framework*. Our operational criteria—shared semantic mechanisms validated via boundary-aware induction—represent a pragmatic balance making large-scale mode discovery possible. Demanding strict causal proof applies an out-of-scope standard ignoring engineering realities. Furthermore, dismissing our definition without proposing concrete, engineering-feasible alternatives makes your requested "substantial changes" non-actionable.
> > >
> > > **2. On Silhouette Scores and Human Evaluation**
> > > You correctly pointed out that silhouette scores in high-dimensional spaces are not conclusive indicators of semantic validity. We agree; however, this critiques a claim we never made. We use silhouette scores solely to verify *structural separability*. Semantic validity is established through a rigorous, **multi-step validation**:
> > >
> > > * **Failure-Aware Embeddings:** Our space explicitly encodes error signals; distance is tightly coupled with failure mechanisms.
> > > * **Boundary-Aware Induction:** Algorithmically ensures clusters represent mechanistic splits.
> > > * **The Intruder Test (Section E3):** We must strictly correct the claim that this verifies mere "semantic coherence." We specifically *control* for semantic similarity by matching intruder and target cases by topic difficulty. **If annotators relied solely on semantic coherence as you claim, they would fail the test completely.** They succeed *only* by isolating the true, underlying error mechanism.
> > >
> > > Dismissing this holistic evaluation—without providing actionable, alternative validation metrics—makes it methodologically impossible for us to optimize against unspecified standards.
> > >
> > > **3. On Presentation Constraints vs. Empirical Depth**
> > > In your initial review, you noted that deferring details to the appendix gave the *impression* of limited depth. We must respectfully disagree with the conclusion. To present a comprehensive validation across 9 RQs and multiple LLMs, we faced agonizing trade-offs under the strict 9-page limit. Deferring supporting analyses to the appendix was a painful necessity to comply with space constraints, not a reflection of insufficient analysis. We have fully committed to restructuring the main text to elevate these details, and we earnestly hope that our efforts to comply with strict formatting limits are not penalized as a lack of empirical rigor.
> > >
> > > **4. Empirical Reality vs. "Conceptual Claims" (Downstream Validation)**
> > > We must respectfully disagree that our mode-level discovery's value relies on "largely conceptual claims" without concrete downstream advantages.
> > >
> > > First, the entire discovery process is empirically grounded across **5 distinct downstream datasets** encompassing various tasks. The discovered modes are direct reflections of downstream performance, not abstract concepts.
> > >
> > > Second, **RQ8 and Figure 7** provide concrete, quantitative measurements of how specific failure modes evolve across model generations (e.g., Llama-2 to Llama-3). Tracking evolutionary trajectories is a highly empirical downstream application that is **structurally unachievable** through traditional case-level discovery. Releasing the ProbeLLM framework as an open-source toolkit further grounds our contribution in immediate practical utility.
> > >
> > > **5. The Requested IFEval Experiments**
> > > Finally, our rebuttal included the **IFEval experiments** exactly as requested in your initial review, demonstrating robust 82%–94% error detection rates across six modern LLMs. While unacknowledged in your final assessment, we trust the Area Chairs will consider these results as fulfilling your task diversity concern.
> > >
> > > We express our deepest gratitude for your rigorous engagement. Your challenging questions pushed us to articulate the boundaries and practical positioning of ProbeLLM with greater precision. We genuinely hope these final clarifications—particularly regarding our pragmatic scope and concrete downstream validations—encourage you to view our contributions in a new light and reconsider your assessment. Regardless of the outcome, we sincerely appreciate the immense effort invested in reviewing our paper. Thank you!

---

### Official Review · Reviewer_7oJp · 2026-03-16

**Soundness:** 3
**Presentation:** 4
**Significance:** 4
**Originality:** 3
**Overall Recommendation:** 6
**Confidence:** 4

**Summary:**

The paper introduces PROBELLM, a framework for automating the diagnosis of LLM failures. It aims to elevate model evaluation and weakness discovery from isolated failure cases to structured failure modes, addressing the challenges that existing static benchmarks are difficult for static evaluations to surface emerging weaknesses with model evolving.

PROBELLM offers a benchmark-agnostic solution by following core components: 1) PROBELLM formulates failure discovery as a hierarchical Monte Carlo Tree Search (MCTS) process, by balancing global exploration and local refinement. 2) PROBELLM incoporated Failure Mode Synthesis, converting individual failure cases (\(F_T\)) into diagnostically meaningful, structured patterns of recurring errors.

**Compliance With Llm Reviewing Policy:**

Affirmed.

**Key Questions For Authors:**

No other questions

**Limitations:**

No siginificant limilations

**Strengths And Weaknesses:**

Strength:
1. The paper provides theoretical proofs in Appendix E for the feasibility of the MACRO-MICRO probing strategy (\(E.1\)) and analyzes the regret bounds of the framework (\(E.2\)). These analyses, based on standard theories from MCTS and multi-armed bandit problems (such as UCB1 and UCT algorithms), demonstrate PROBELLM's asymptotic optimality in failure discovery, providing a solid mathematical basis for the framework's design.
2. The paper is clearly written, well-structured, with detailed and thorough experiments.
3. The paper addresses a highly important and urgent problem in the LLM domain, and its contributions have broad implications. and presents a novel and effective framework for LLM failure diagnosis.

Weakness:
1. Even though the framework itself is designed to be independent of specific benchmarks during its probing phase, the results would be influence by the initial seeds

---

> ### Author Rebuttal · Authors · 2026-03-28
>
> Thank you for your supportive review and recognition of our work. We address your comments below.
>
> ### W1: Results influenced by initial seeds
>
> We understand your concern. ProbeLLM is explicitly designed to mitigate seed bias:
>
> - **MACRO exploration** generates samples **outside existing embedding clusters**, encouraging exploration beyond seed regions.
> - **Adaptive budget allocation (via MCTS)** continuously reallocates effort toward under-explored but high-yield regions, preventing long-term bias from initialization.
> - **Diverse generation mechanisms** (e.g., tool-augmented reasoning and perturbations) further reduce dependence on seed semantics.
>
> Empirically, our **cluster-overlap analysis (RQ4)** shows that ProbeLLM discovers a substantial number of **novel failure modes beyond seed benchmarks**, and the increasing cluster count (RQ1) further indicates expansion beyond initial regions.
>
> Overall, ProbeLLM expands the failure landscape rather than merely refining seed-induced regions.
>
> Thank you again for your strong endorsement and constructive suggestion!

---

### Decision · Program_Chairs · 2026-04-30

**Decision:**

Accept (regular)

**Comment:**

Reviewers appreciated the paper's shift from case-level to mode-level failure discovery, its principled MCTS formulation, and its broad empirical validation across 12 models with mitigation and human evaluation. One reviewer is weakly advocating for rejection, most notably arguing that "failure mode" framing implies mechanistic structure while the evidence only establishes coherent clustering with LLM-generated labels. This feels to me like a revision-level framing issue, not a fundamental flaw, and remaining reviewers all advocate for acceptance, including one strong accept.